# BODIPY-Based Nanomaterials—Sensing and Biomedical Applications

Tomasz Koczorowski [1,*], Arleta Glowacka-Sobotta [2], Stepan Sysak [1,3], Dariusz T. Mlynarczyk [1],
Roman Lesyk [4,5], Tomasz Goslinski [1] and Lukasz Sobotta [6]

[1] Chair and Department of Chemical Technology of Drugs, Poznan University of Medical Sciences, Grunwaldzka 6, 60-780 Poznan, Poland; stepan.sysak@student.ump.edu.pl (S.S.); mlynarczykd@ump.edu.pl (D.T.M.); tomasz.goslinski@ump.edu.pl (T.G.)

[2] Chair and Department of Maxillofacial Orthopedics and Orthodontics, Poznan University of Medical Sciences, Bukowska 70, 60-812 Poznan, Poland; aglow@ump.edu.pl

[3] Doctoral School, Poznan University of Medical Sciences, Bukowska 70, 60-781 Poznan, Poland

[4] Department of Biotechnology and Cell Biology, Medical College, University of Information Technology and Management in Rzeszow, Sucharskiego 2, 35-225 Rzeszow, Poland; roman.lesyk@gmail.com

[5] Department of Pharmaceutical, Organic and Bioorganic Chemistry, Danylo Halytsky Lviv National Medical University, Pekarska 69, 79010 Lviv, Ukraine

[6] Chair and Department of Inorganic and Analytical Chemistry, Poznan University of Medical Sciences, Rokietnicka 3, 60-806 Poznan, Poland; lsobotta@ump.edu.pl

* Correspondence: tkoczorowski@ump.edu.pl

**Abstract:** Cancerous diseases are rightfully considered among the most lethal, which have a consistently negative effect when considering official statistics in regular health reports around the globe. Nowadays, metallic nanoparticles can be potentially applied in medicine as active pharmaceuticals, adjustable carriers, or distinctive enhancers of physicochemical properties if combined with other drugs. Boron dipyrromethene (BODIPY) molecules have been considered for future applications in theranostics in the oncology field, thus expanding the potential of conceivable applicability. Hence, taking into account positive practical features of both metal-based nanostructures and BODIPY derivatives, the present study aims to gather recent results connected to BODIPY-conjugated metallic nanoparticles. This is with respect to their expediency in the diagnosis and treatment of tumor ailments as well as in sensing of heavy metals. To fulfill the designated objectives, multiple research documents were analyzed concerning the latest discoveries within the scope of BODIPY-based nanomaterials with particular emphasis on their utilization for diagnostical sensing as well as cancer diagnostics and therapy. In addition, collected examples of mentioned conjugates were presented in order to draw the attention of the scientific community to their practical applications, elucidate the topic in a consistent manner, and inspire fellow researchers for new findings.

**Keywords:** BODIPY; cancer; fluorescence; nanoparticles; photosensitizer

## 1. Introduction

### 1.1. Nanotechnology and Nanomaterials in Medicine

Miniaturization processes of bulk materials reduced to the nanoscale have led to the development of structures with interesting physicochemical properties up to 100 nm in size, called nanoparticles. Nanotechnology brings many opportunities for the application of nanoparticles in numerous fields, including medicine and pharmacy, biotechnology, electronics, optics, and even automotive and construction industries [1–6].

Nanotechnology conducted in life sciences is defined as nanobiotechnology and its benefits are further displayed in the field of medicine and pharmacy, especially in the early diagnosis of diseases and the improvement of treatment methods. Nanotechnology has impacted the development of nanomedicine, especially in terms of drug delivery approaches for potential use in therapy and diagnostics [7]. Nanomedicine is now considered

in many medical applications: (i) diagnostics, (ii) imaging techniques, (iii) biomaterials for internal and external use, (iv) new pharmaceuticals and drug delivery systems for targeted therapies, and (v) clinical and toxicological aspects of nanodrugs utilization (Figure 1) [8].

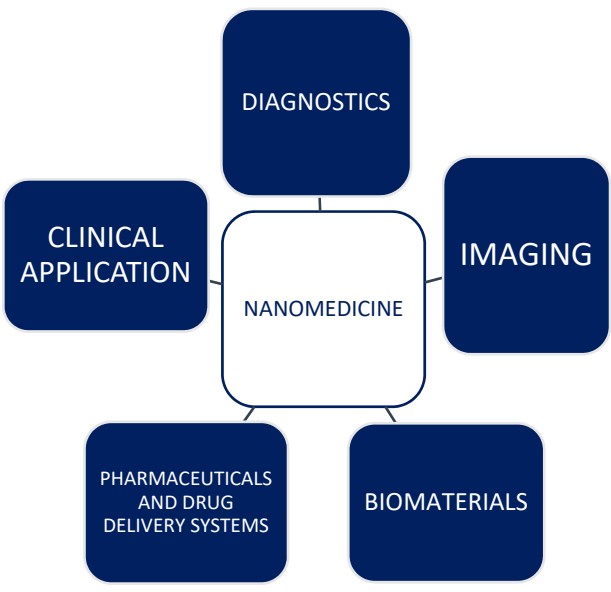

**Figure 1.** The nanomedicine scope of interest [8].

Progress in nanotechnology is strictly related to new discoveries in the nanomaterials field. According to European Commission guidelines, the term "nanomaterial" means both natural and artificial material where, for at least 50% or more of all particles, one or more external dimensions is in the size range 1–100 nm [9]. Nanoscale dimensions result in special physicochemical as well as optical features of nanomaterials. Nanomaterials are known for their high surface-area-to-volume ratio, which allows a vast number of molecules to be attached to their surface [10]. Another significant property is the quantum size effect, which allows nanomaterials to display individual electronic band structures as molecules on the macroscale [10]. In addition to this, nanomaterials also reveal some other interesting features: (i) increased molecular adsorption and surface tension force, (ii) enhanced both biological and chemical (catalytic) activities, (iii) fast clustering tendency, (iv) lower melting points, and (v) significant mechanical strengths (Figure 2) [10,11]. However, for most applications, raw nanoparticles have to be physically or chemically modified before being used [12].

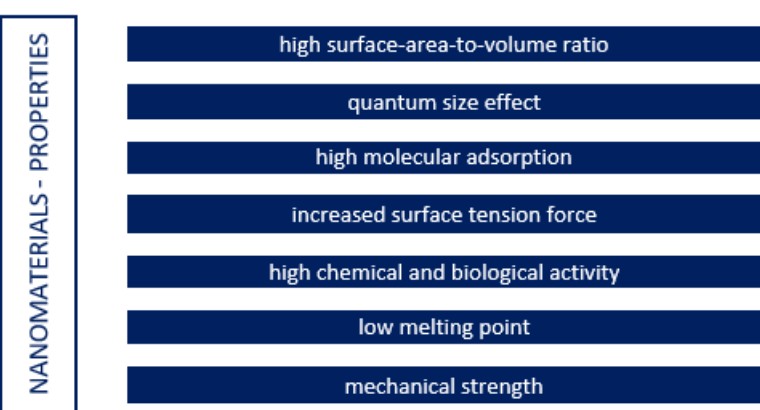

**Figure 2.** Unique features of nanomaterials [10,11].

### 1.2. Metallic Nanoparticles in Medicine

Among numerous types of nanomaterials, metallic nanoparticles are one of the most interesting ones. The widely spread method of obtaining metal-based nanoparticles is the chemical reduction of the metal compound (e.g., chloroauric acid or silver nitrate) via a proper reducing agent (e.g., sodium citrate or sodium borohydride), in some cases also used as a stabilizer allowing the prevention of unwelcome agglomeration [13]. The diameter of metallic nanoparticles can be easily tuned as reaction parameters change, including time of reaction, temperature, amount of precursor, and reducing agent. Nowadays, the synthesis of nanoparticles also involves green chemistry methods to reduce the usage of environmentally harmful reagents [13,14]. In this way, some reducing agents such as green sources are used, e.g., lactic acid, citrus fruits, coffee seeds, etc. In addition, sonochemical, microwave-assisted, and electrochemical methods are also involved in the production of metallic nanoparticles [15].

The unique properties of metallic nanoparticles enable their use for various purposes in nanomedicine. One of the most important fields is tumor imaging, where metallic nanoparticles form a class of new contrast and tracking agents [16]. Due to their structures, they are suitable for detection, diagnosis, and treatment of various tumor lesions. Metallic nanoparticles can also be a part of the theranostic nanoconcept as a combination of both diagnosis and treatment [16]. Utilized as drug delivery carriers, metallic nanoparticles are solid colloidal particles with diameters from 10 to 1000 nm [17]. There are several possibilities to bind a therapeutic agent with its nanocarrier, such as (i) encapsulation within a polymer shell or a structure, (ii) dispersion in a polymer carrier matrix, and (iii) covalent attachment or adsorption to the particle surface [18]. Due to site-specificity, prevention of multidrug resistance, and effective delivery of therapeutic agents (based on targeting ligands), metallic nanoparticles are able to increase the therapeutic index of specified drugs and are now being investigated for cancer treatment. The increased surface-area-to-volume ratio of some metallic nanoparticles allows chemical modification of their surface, which results in an increased cellular uptake, protection of the therapeutic agents, and in some cases even an improvement of the bioavailability of the anticancer agents [19]. Nanocarriers, both alone and with modified surfaces, have many interesting features. They reveal protective effects towards various drug molecules in different compartments, which can increase their stability and minimize eradication. Nanometer size range carriers can also participate in the enhanced permeability and retention (EPR) effect, which is important for medical applications. What is more, they can also be involved in multiple drug deliveries and various combined therapies, e.g., by utilizing chemotherapeutic and photothermal effects (Figure 3) [20].

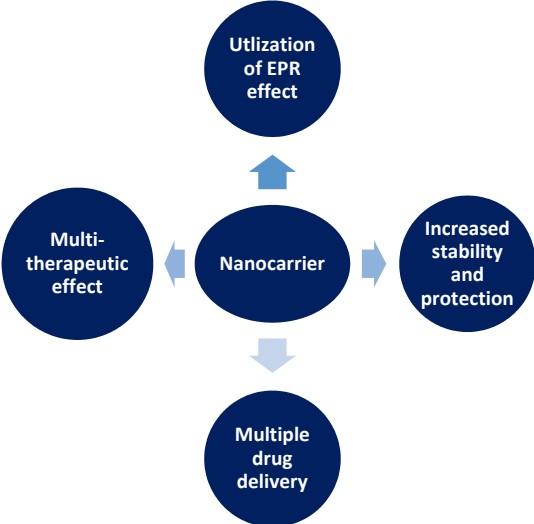

**Figure 3.** Features of nanocarriers for free drug administration [20].

*1.3. Brief Characterization of Selected Nanoparticles*

The selected nanoparticles below were introduced and discussed in terms of the most important physicochemical features of potential applicability in nanomedicine.

There is no doubt that gold nanoparticles are among the most frequently applied metallic nanoparticles for various purposes, especially in medicine. They are produced via the reduction reaction of gold salts and are stabilized by phosphines, alkanethiols or citrates. The surface of gold nanoparticles can easily be chemically modified with ligands containing functional groups such as thiols, phosphines, and amines [21]. The functional groups of the surface ligands allow some other moieties to be attached, such as oligonucleotides, proteins, and antibodies, to modify the functionality of such hybrids. For example, coating of Ag nanoparticles with thiolated PEG or incorporation into liposomes can lead to species that can evade the opsonization and bypass the immune system. Gold nanoparticles can be widely used in nanomedicine, particularly in cancer therapy [22,23]. Gold nanoparticles reveal a significant feature-surface plasmon resonance, which means the ability to absorb and scatter light of wavelengths considerably larger than the size of a particle. The treatment method that employs plasmon resonance is photothermal therapy (PTT), where irradiation with light from the wavelength range of 800–1200 nm causes heat release resulting in tumor cells destruction, which is based on protein denaturation and damage of nucleic acids and cell membrane, as well as the generation of reactive oxygen species (ROS) [24]. As drug carriers, gold nanoparticles can be utilized as passive and/or active transporters. When they act as passive carriers, their absorption and accumulation are based on the EPR effect. As active carriers, gold nanoparticles can be combined with antibodies providing their desired localization [23]. However, application of gold nanoparticles in the treatment can lead to some toxic effects, depending on their size and surface modification. As mentioned above, apart from the heat release, when irradiated with light or ultrasounds, gold nanoparticles can also generate cytotoxic ROS. For this reason, gold nanoparticles can also be considered as photo- and sonosensitizers for photodynamic therapy (PDT) and sonodynamic therapy (SDT), respectively [23,25–27]. Moreover, conjugation of gold nanoparticles with photosensitizers can significantly increase the concentration of photosensitizers in tumor tissue in comparison to the accumulation of photosensitizer alone. What is more, after light irradiation, the energy transfer from photoexcited electrons is elevated between photosensitizer (PS) and gold nanoparticles [27].

Frequent usage of antibiotics results in an increase in drug resistance among bacteria and forces the need to search for alternative sources for antimicrobial treatment. Silver nanoparticles (AgNPs) seem to meet these requirements as they were developed as promising, efficacious inhibitors of microorganisms' growth with potential medical applications in ophthalmology, dermatology, orthopedics, and dentistry [13,28,29]. Similar to gold nanoparticles, silver nanoparticles can be synthesized by reducing silver salts. Furthermore, the silver core is usually coated with polyvinylpyrrolidone (PVP) or citrate to increase its stability [13]. The antimicrobial effect of the AgNPs is based on many mechanisms: (i) they attach to the cell wall of bacteria or fungi and cause damage to the structure of the cell membrane and leakage of intracellular components, (ii) they bind to DNA, thus hampering replication, and finally, (iii) silver ions can produce radicals that can decrease the activity of enzymes [28]. The main obstacle to the development of antimicrobial effects of silver nanoparticles is the determination of proper concentration for selected pathogens and the establishment of treatment algorithms [29]. In addition to the antimicrobial effect, silver nanoparticles also exhibit radiosensitizing properties. In this case, the mechanism of action relies on the induction of apoptosis, activation of oxidative stress, and increase in the permeability of the cell membrane [17].

Magnetic nanoparticles constitute another type of interesting nanomaterial. Their core, formed of $Fe_3O_4$ or $\gamma$-$Fe_2O_3$, can be coated with polyvinyl alcohol (PVA), dextran, PEG, PVP, or chitosan to protect them from rapid clearance within the blood system. The synthesis of magnetic nanoparticles is based on alkaline coprecipitation of iron(II) or iron(III) ions [30]. Thanks to their supermagnetic properties, these nanoparticles can be transported to the

desired location when an external magnetic field is applied. In addition, they can also release heat at their location, causing cell death. Similar to other metallic nanoparticles, the toxicity of the magnetic nanoparticles is also under investigation. The reaction of iron ions with hydrogen peroxide or oxygen generates ROS, leading to DNA damage. Despite this, magnetic nanoparticles are considered as attractive drug carriers due to their high loading capacity and the possibility of surface modification [31]. Nowadays, they are also widely used in nanomedicine and nanobiotechnology as biological sensors and contrast agents in magnetic resonance imaging.

Quantum dots (QDs) belong to interesting metallic and semiconducting nanostructures with a size range of 2–10 nm and are made of inorganic nanocrystals usually composed of a CdSe core or ZnS shell and coated with PEG [32]. QDs are photostable and possess broader excitation spectra in comparison to narrow excitation-emission spectra used in imaging techniques. Moreover, their excited state after light irradiation is prolonged [33]. They are often encapsulated in micelles or attached to amphiphilic molecules and polysaccharides to provide good water solubility. The quantum dots can penetrate tissues well due to their small size and the EPR effect. QDs conjugated with tumor-specific antibodies can be applied in imaging techniques in oncology. Their fluorescence properties allow only one light source to be used, which results in lower costs and easier data analysis. Moreover, in imaging with QDs, no signal amplification is needed [34]. QDs can also be applied in theranostics as both drug carriers and imaging agents. Despite this, there are some controversies about the use of QDs in medicine due to their potential toxicity related to the release of toxic cations. QDs can also act as energy donors and acceptors in fluorescence resonance energy transfer (FRET) when conjugated with photosensitizers [35].

Titanium dioxide ($TiO_2$) also belongs to the group of semiconducting metallic nanoparticles. This naturally occurring mineral is commonly used as a white pigment. Its utilization in medicine or environmental protection is based on its photocatalytic properties [36]. After excitation of $TiO_2$ with UVA light, electron emission is observed, accompanied by a positively charged hole left behind in the molecule. The free electron can react with oxygen to generate superoxide, whereas the hole can produce hydroxyl radicals from water [27]. The production of ROS via irradiation of $TiO_2$ can be used for photocatalytic an antimicrobial effect or degradation of organic pollutants. Nowadays, $TiO_2$ nanoparticles are also used as photosensitizing agents in hybrid materials [37].

Upconverting nanoparticles (UCNs) belong to modern metallic nanoparticles, which after sequential absorption of two or more photons can emit light at shorter wavelengths in comparison to the excitation wavelength [27]. UCNs can convert low energy light (near-infrared wavelengths) to high energy light through the anti-Stokes emission process. Materials exhibiting such photon upconversion are usually comprised of host ceramic lattices embedded with a transition metal, as well as actinide or lanthanide ions such as $Yb^{3+}$, $Er^{3+}$, and $Tm^{3+}$. Using near-infrared wavelengths for excitation of UCNs allows for deeper tissue penetration. The emitted shorter wavelengths can further excite other PSs attached to the lanthanide-doped nanoparticles [38].

Another compelling group of metallic nanomaterials are zinc oxide (ZnO) nanoparticles, which are nowadays used as light-emitting diodes, photocatalysts, and biosensors [39]. Like silver, ZnO also exhibits antibacterial activities and can be considered a safe and stable inorganic antimicrobial agent [40]. Chemically synthesized from zinc sulfate, ZnO nanoparticles can cause cell membrane damage via ROS production, which occurs on their surface. It was proved that ZnO nanoparticles could provide significant growth inhibition within a wide range of pathogenic bacteria after irradiation of visible light [40].

Recently, metal-organic frameworks (MOFs) have been proposed in nanomedicine as both promising drug delivery carriers and theranostic agents. These nanostructures are formed via the reticular synthesis of metal-containing subunits (inorganic clusters) with organic linkers (carboxylates, imidazolates, and phosphonates) to give open crystalline frameworks with high flexibility and porosity [41,42]. A vast number of available metal ions and organic linkers accompanied by a diversity of structural motifs leads to countless

combinations in the design of MOFs [43]. Postsynthetic modification of MOFs allows for conjugation with various biologically active molecules for various purposes in medicine. They are known for their large surface area and high porosity, which allows them to host guest structures [43]. MOFs have recently been considered potential gas and vapor-phase sensors regarding their porosity. In the last decade, they have also been connected with porphyrinoids and related compounds for chemical sensing and catalysis [44,45].

To sum up:

- Some nanoparticles, such as gold and $TiO_2$, can generate ROS species when irradiated with light at the proper wavelength.
- Silver and zinc oxide nanoparticles reveal antibacterial activity.
- The most interesting feature of quantum dots is their fluorescence activity.
- The emission of light of shorter wavelength after low-energy light excitation is an attribute of upconverted nanoparticles.
- MOFs can be used as potential drug delivery systems due to their large surface area and high porosity.

*1.4. Nanoparticles Involved in the Photodynamic Therapy*

The conjugation between metallic nanoparticles and porphyrinoid-type photosensitizers is an interesting issue for medical, photocatalytic, and sensing purposes. Metallic nanoparticles conjugated with porphyrinoids are of great interest for medical applications since they could overcome the limitations of classic PS used in PDT. Nanoparticles can be categorized into passive and active nanoparticles depending on their involvement in photosensitizer excitation. Nanoparticles as PS carriers offer some benefits, such as the hydrophilicity of hybrid material and passive targeting of tumor tissues via the EPR effect. Direct targeting of nanoparticles to specific tissues can be reached via nanoparticles' surface functionalization with monoclonal antibodies or specific tumor-targeting molecules (proteins, peptides, and aptamers) [39,46]. Unlike typical drug delivery systems, the release of PS by metallic nanoparticles is not required [47].

In PDT, gold nanoparticles, due to their localized surface plasmon resonance, could be used to increase the excitation efficiency of the accompanying PS [48]. Another type of nanoparticle which is useful in PDT is the group known as QDs, the surface of which can be modified and functionalized to make it water-soluble and more biocompatible. Due to the large transition of the dipole moment, they are strong light absorbers and can act as energy donors and transfer energy, finally leading to the generation of ROS. The transfer of energy from QD to PS via FRET can also be potentially applied in PDT due to deeper tissue penetration of near-infrared (NIR) light irradiation [35]. Apart from QDs, UCNs can also be used in photodynamic reactions to convert NIR excitation to lower energies, which are capable of activating the accompanied PS. What is more, the surface of $TiO_2$ nanoparticles can be modified with metal porphyrins using impregnation based on simple incubation of the synthesized $TiO_2$ nanoparticle powder in an organic solution of metal porphyrins. The adsorption of metal porphyrins on the $TiO_2$ surface can be realized in different ways, including (i) chemical functionalization, (ii) axial coordination of bridging ligands with metal atoms, (iii) intermolecular $\pi$–$\pi$ interactions between planar porphyrin surfaces, and (iv) intermolecular hydrogen bonds due to the introduction of functional groups in porphyrin molecules [49]. Via the conjugation of metal porphyrinoids with $TiO_2$, the absorption of visible light also becomes possible [50].

Metallic nanoparticles in conjunction with PSs reveal many advantages for their prospective use in PDT: (i) the amount of PS around tumor tissue can be increased due to a large surface-to-volume ratio of nanoparticles, (ii) nanoparticles may prevent tissue from fast elimination and inactivation of PSs via plasma components, (iii) nanoparticles may prevent normal tissues from nonspecific accumulation of PSs and reduce their overall photosensitivity, (iv) the amphiphilicity of PS can be obtained via conjugation with nanoparticles, allowing them to be distributed in the cardiovascular system and be further localized in the tumor tissue, (v) nanocomposites delivery and retention in tumor tissue is combined

with the EPR effect, (vi) the surface modification of nanoparticles with functional groups or targeting agents can improve their biodistribution, pharmacokinetics, cellular uptake, and targeting abilities, and (vii) nanoparticles allow the carrying of multiple components to the target tissue; specifically imaging agents, drugs, targeting ligands, and agents preventing interference with the immune system (Figure 4) [39].

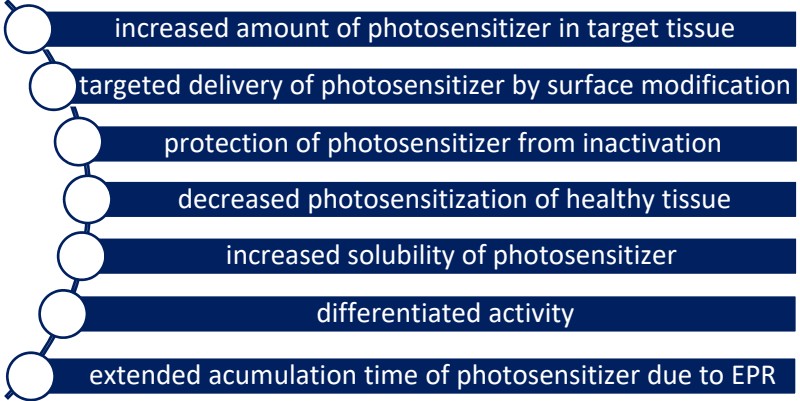

**Figure 4.** Advantages of photosensitizer–nanoparticle conjugation [39].

To sum up:

- In PDT, the nanoparticles can be used as passive drug carriers or active co-photosensitizers.
- To the most frequently studied nanoparticles for PDT belong gold nanoparticles, quantum dots, upconverted nanoparticles, and titanium dioxide nanoparticles.
- To the advantages of combining nanoparticles and photosensitizers in PDT belong increased nanocarrier surface, better solubility of PS, and better permeability of nanoparticles due to the EPR effect and dual mechanism of action.

*1.5. BODIPY Dyes*

Dipyrromethene boron difluorides (BODIPYs) constitute an essential part of modern diagnostic and healing approaches. While enclosing rich physicochemical features, these fluorescent dyes found their applications in multiple fields, including medicine, pharmacy, and environmental sciences. In various in vitro studies, BODIPY derivatives have been successfully used for labeling in evaluating drugs' efficacy, localization, distribution, detection, and mechanism of action [51–53]. They were also used as contrast agents in photoacoustic imaging (PAI) to improve in vivo imaging techniques, playing a huge role in providing cancer cells' location and metabolic activity [54]. The conjugation of BODIPYs with metallic nanoparticles has proved to be an effective practice that increases the dyes' properties.

The derivatives of BODIPY have been studied for many reasons (Figure 5), of which the improvement of the permeability of cell membranes for nanomedications and their use in PDT are particularly important [27]. BODIPY-functionalized gold nanoparticles can be considered as PSs in antimicrobial PDT [55]. In addition, they may also act as optical sensors and play an essential role in cell imaging by enhancing the fluorescent signal from confocal microscopy and flow cytometry [56]. Peptide-functionalized gold nanoparticles with BODIPY have revealed a significant role in the controlled drug distribution in the future.

There is a vast interest in the development of fluorescent chemosensors for the recognition of environmentally important analytes. It seems that environmental protection can be improved with BODIPY derivatives, as they can be applied to detect toxic metal ions, such as $Cr^{6+}$, $Hg^{2+}$, $Ag^+$, $Cu^{2+}$, and $Cd^{2+}$ in aqueous solutions. So far, BODIPY-based dual-channel chemodosimeters have been developed for physiologically harmful ions, such as mercury and silver [57]. BODIPY-functionalized magnetite fluorescent nanocomposites are of great interest, as they may be a new class of non-toxic sensors for environmental applications [58]. These improved analytical methods for monitoring environmentally-unfriendly metals could be beneficial due to growing water pollution.

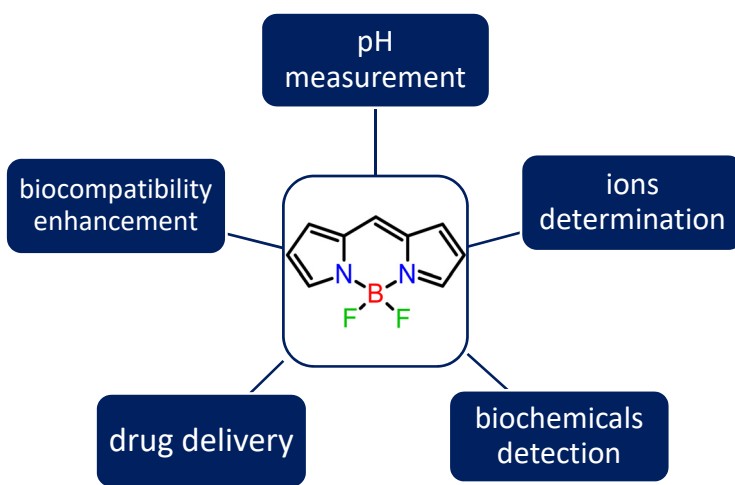

**Figure 5.** Metallic nanoparticles/BODIPY in medicine [51–58].

In this paper, we have focused on BODIPY derivatives in conjugation with gold and silver nanoparticles, magnetic, ZnO, TiO$_2$ nanoparticles, MOFs, and UCNs. The review presents examples of such species that have been discussed in the scientific literature in the last decade, emphasizing their potential applications in nanomedicine, PDT, PACT, PTT, sensing of biologically relevant molecules, and tumor tissue imaging.

To sum up:

- Photosensitizers based on BODIPY dyes can be considered for photodynamic therapy and antimicrobial treatment as well as for the treatment of skin surface lesions.
- BODIPY dyes reveal sensing properties towards the determination of heavy metal ions.

## 2. Metallic Nanoparticles@BODIPY

For clarity, this manuscript chapter was divided into several subchapters regarding the type of metallic nanoparticle used as a BODIPY carrier.

### 2.1. Sensing Applications

#### 2.1.1. Magnetite Nanoparticles

There are many drugs with a very narrow "therapeutic window". This means that some drugs act in the desired way only in a narrow concentration range, which means that at a concentration below this range, there is no curing effect, and above it, a toxic one is observed. An example of these substances is tacrolimus—an immunosuppressive drug crucial in the transplantation field with therapeutic concentrations from 5 to 20 ng/mL. Tacrolimus is strongly bound to proteins; there is only 0.5% of the free fraction. Therefore, a fast and straightforward quantification method is needed to provide a safe therapy. Salis and co-workers proposed the determination of this drug based on fluorescent magnetic nanobeads composed of BODIPY dyes as a fluorescent probe [59]. BODIPY dyes were deposited on polystyrene nanospheres antecedently decorated with ferrite nanograins and exposed to carboxylic functionalization (Figure 6). Later, the activated particles were conjugated to capture IgG anti-IgM antibodies through the amine groups. The procedure of tacrolimus determination relies on the addition of anti-FK506 (anti-tacrolimus) and anti-biotin antibodies to the sample with carboxylated drugs deposited on a glass surface where they competitively bind each other. After rinsing the sample with PBSTM buffer, nanoparticles were added to it and interacted with tied antibodies. The interaction was accelerated by a Nd magnet that kept the particles in the close neighborhood of tacrolimus-antibody glass. Finally, the fluorescence of fettered particles was measured. Thanks to the FRET phenomenon and significant Stokes shift, the brown color of ferrite did not disturb the measurement. The authors determined the tacrolimus detection limit to be 0.08 ng/mL [59].

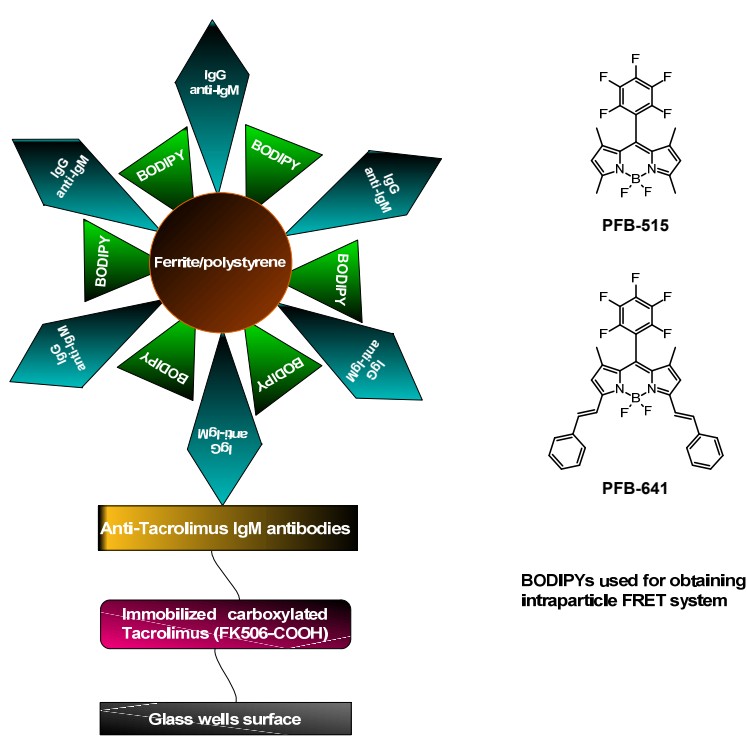

**Figure 6.** Schematic representation of the conjugates based on BODIPY dyes and antibodies deposited on ferrite/polystyrene nanosheets [59].

In another study, Lee and co-workers designed magnetic silica nanoparticles based on BODIPY dye with nickel and applied them for selective detection and removal of lead ions both from the blood and water environment (Figure 7) [60]. Lead is well-known for causing dangerous side effects in humans. This element is responsible for muscle paralysis, neuropathy, and anemia. The authors prepared core/shell Ni@SiO$_2$ particles and attached BODIPY dye to them via a sol-gel reaction. Obtained particles bearing bounded Pb$^{2+}$ ions were spherical at a narrow size range of 30–40 nm. Nanoprobes without lead ions did not reveal fluorescence ability, which was expected as a result of efficient photoinduced electron transfer—PET. Continuous binding of lead ions turns on fluorescence, according to the chelation-enhanced fluorescence effect, which blocks the PET phenomenon. The addition of 15 ppb lead ions, which constitutes the acceptable level of lead in drinking water, caused over a 10% increase of fluorescence quantum yield, and this heavy metal was removed in 97%. Lead bounding by nanoparticles is reversible, and particles are active after a few detection/separation cycles. Moreover, the authors performed studies to determine the selectivity of lead ions in the presence of Li$^+$, Na$^+$, Mg$^{2+}$, K$^+$, Ca$^{2+}$, Cu$^{2+}$, Zn$^{2+}$, Ag$^+$, Cd$^{2+}$, and Hg$^{2+}$, and noticed that t two carbonyl groups forming coordination bonds play an important function in Pb$^{2+}$ binding. Finally, the authors performed experiments concerning the decontamination of a blood sample. In the blood, they placed 100 ppb of Pb$^{2+}$—the amount believed as unsafe and noted that the designed nanoparticles efficiently removed 96% of the total lead content [60].

Son and co-workers proposed a similar nanoprobe for lead ions detection [61]. They used the same BODIPY molecule as Lee and co-workers and core/shell Fe$_3$O$_4$@SiO$_4$ nanoparticles components (Figure 8). The detection limit was determined at 1.5 ppb, and the response time of 60 s was noted. The authors confirmed the selectivity of the nanoprobe and its reproducibility within a few cycles of lead detection and further sensor stripping with EDTA. Moreover, the fluorescence signal was unchanged in the pH range of 3–11. Thus, they concluded that their nanoprobe might be used in the physiological environment. The usability of the lead nanodetector was checked in the living cells. Pb(ClO$_4$)$_2$ and a nanoprobe were added to the HeLa cells, and then the fluorescence signal was measured.

The authors found that cells treated with lead ions showed 10-fold higher emission intensity in comparison to free-lead cells. Moreover, the sensor was active in many cycles of lead detection stripping [61].

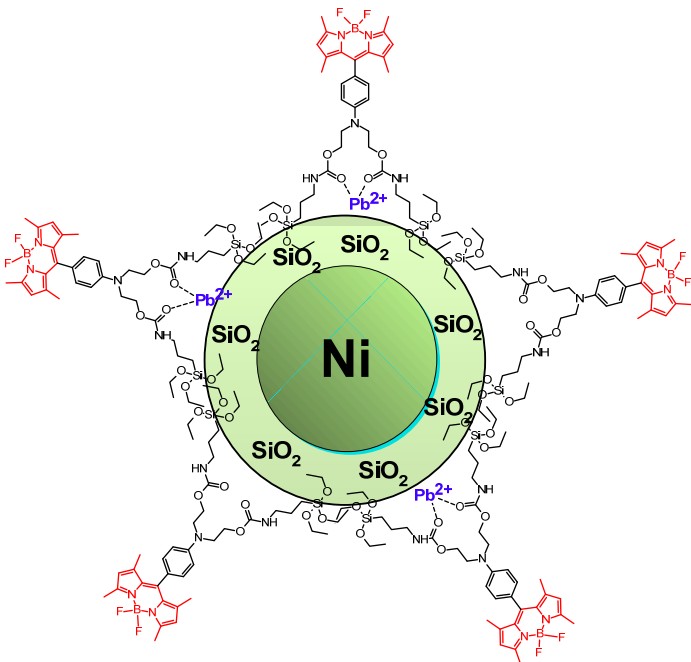

**Figure 7.** The chemical structure of nickel nanoparticles coated with silica and BODIPY dyes [60].

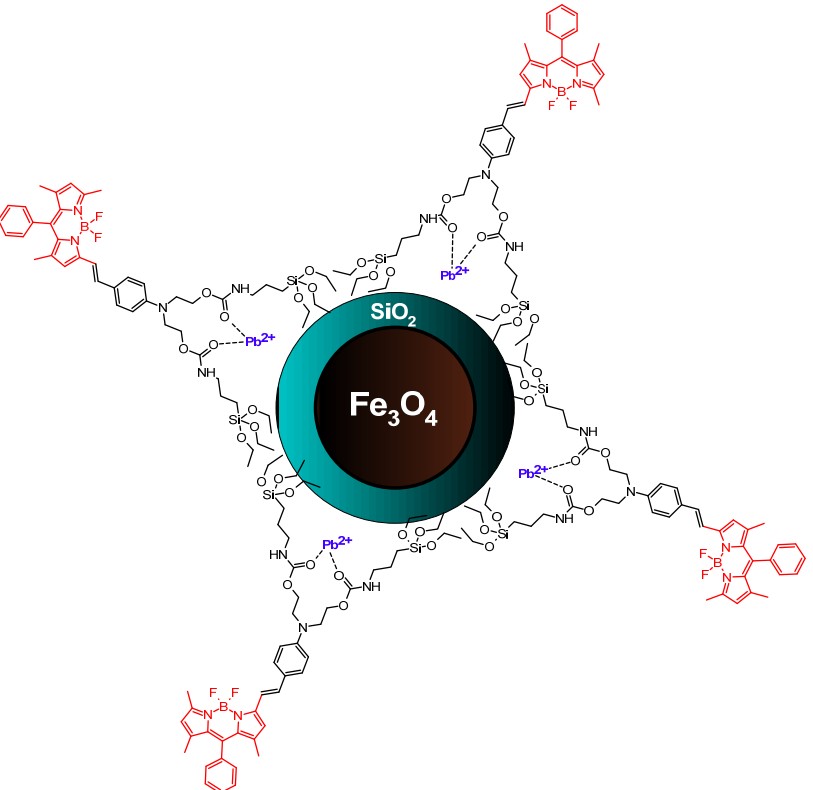

**Figure 8.** The chemical structure of magnetite nanoparticles coated with silica and BODIPY dyes [61].

Heavy metal pollution represents a considerably severe ecological issue and can invoke numerous disorders in the human organism, including cancer. In 2020, Bilgic and Cimen de-

signed a new fluorescent superparamagnetic nanoparticle (F-SPION) that revealed a selective fluorescent response against ions $Cr^{6+}$ in the strongly acidic medium [62,63]. F-SPION was derived from superparamagnetic magnetite particles coated with silica and modified by 3-aminopropyltrimethoxysilanes (APTMS) bound to *p*-tolyl BODIPY moieties (Figure 9). According to the conducted X-ray diffraction analysis, F-SPION is 18 nm in diameter, while fluorescence spectroscopy revealed its single emission at 510 nm. The fluorescence was also measured in the presence of other cations ($Ag^+$, $Pb^{2+}$, $Mn^{2+}$, $K^+$, $Ba^{2+}$, $Mg^{2+}$, $Cr^{3+}$, $Cd^{2+}$, $Ca^{2+}$, $Hg^{2+}$, $Cu^{2+}$, $Al^{3+}$, $Fe^{2+}$, $Sr^{2+}$, $Fe^{3+}$, $Zn^{2+}$, $Co^{2+}$, and $Ni^{2+}$) as perchlorate salts, but only Cr(VI) ions provoked notable "ON-OFF" emission changes significantly lowering the fluorescent intensity. Experimental studies on pH were conducted in the range of 1–8 and established the value of 1 as the most suitable level for $Cr^{6+}$ evaluation. The limit of ion detection was determined as 3.3 ($\pm$1) mcg/mL [62]. In the following article, the authors reported the formation of two other BODIPY-containing magnetite-based nanosensors named $Fe_3O_4$@$SiO_2$-TPED-BODIPY and $Fe_3O_4$@$SiO_2$-TMPTA-BODIPY [62]. The new compositions shared a lot of similarities with F-SPION in terms of structure and physicochemical attributes. The key difference was the choice of silane agents during the synthesis process: for $Fe_3O_4$@$SiO_2$-TPED-BODIPY, it was *N*-[3-(trimethoxysilyl)propyl]ethylenediamine (TPED), resulting in 18.5 nm nanospheres, while for $Fe_3O_4$@$SiO_2$-TMPTA-BODIPY $N^1$-(3-trimethoxysilylpropyl)-diethylene-triamine (TMPTA) was used, affording the approximate size of 19 nm. The new nanoparticles endured the same set of physicochemical characterizations and proved to be effective sensing instruments towards Cr(VI) ions determination. Based on the emission changes in the fluorescence intensities, the limit of $Cr^{6+}$ detection was ascertained as 2.2 ($\pm$1) mcg/mL for the TPED-nanosensor and 3.1 ($\pm$2) mcg/mL for the TMPTA-nanosensor (pH = 1). The fluorescence of the free $Fe_3O_4$@$SiO_2$-TPED-BODIPY showed higher intensity (286 nm vs. 253 nm) and demonstrated a 70% drop in response to chromium ions (ligand:metal binding ratio 2:1), while in the case of $Fe_3O_4$@$SiO_2$-TMPTA-BODIPY-Cr(VI), the decrease was 77% (ligand:metal binding ratio 1:1) [62].

As the continuation of the study, the authors expanded the collection of designed fluorescent nanoparticles with three novel entities—SPION@$SiO_2$-N1-BODIPY (N1), SPION@$SiO_2$-N2-BODIPY (N2), and SPION@$SiO_2$-N3-BODIPY (N3) [63]. This time, the researchers focused on the absorbing abilities of created nanoparticles towards eliminating Cr(VI) ions from aqueous solutions. The preparation of the new SPIONs echoes the protocol designed for F-SPION, involving the composition of silica-coated magnetite nanospheres with the subsequent application of aminopropyltrimethoxysilanes (APTMS for SPION@$SiO_2$-N1-BODIPY, TPED for SPION@$SiO_2$-N2-BODIPY, and TMPTA for SPION@$SiO_2$-N3-BODIPY), and the conclusive introduction of *p*-tolyl BODIPY molecules into terminal amino groups. However, the reaction time in the silanization step was reduced from 72 h to 24 to retain free silanol groups of silica coating (FT-IR analysis confirmed their presence). As a result, the particle N1 revealed an average size of 18 nm, N2 had 18.5 nm, and N3 had 19 nm. The authors assessed the maximum Cr(VI) ions absorption capacity at a pH of 2 and room temperature employing potassium chromate solution as a source of $Cr^{6+}$. The following values were noted: 23.98 mg/g (equilibrium contact time 180 min) for N1, 33 mg/g (equilibrium contact time 180 min) for N2, and 43.49 mg/g (equilibrium contact time 150 min) for N3. The authors concluded that the absorbing properties advance with the increase of nitrogen-donating groups in a particle. Thus, based on physicochemical studies, a suitable physisorption mechanism was proposed [63].

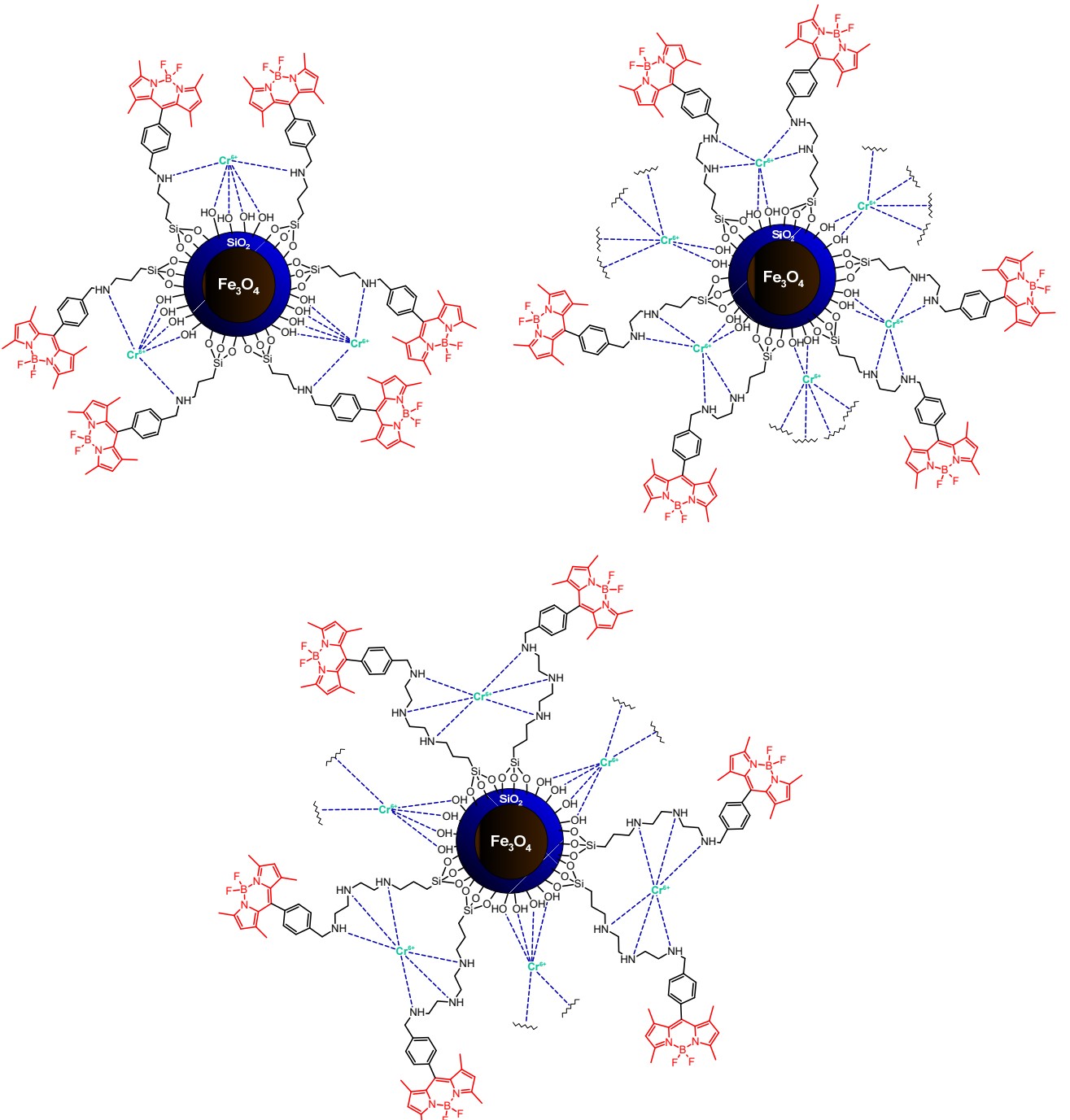

**Figure 9.** The chemical structures of magnetite nanoparticles coated with silica, modified by amino-propyltrimethoxysilanes (APTMS) bound to *p*-tolyl BODIPY moieties [62].

### 2.1.2. Gold Nanoparticles

Lee and co-workers continued the above-presented research for new nanoprobes [64]. They developed a nanosensor based on gold nanoparticles and BODIPY dyes to detect $Cu^{2+}$ ions (Figure 10). Obtained nanoparticles also revealed a spherical shape at a small size equal to 7–10 nm. Quenching of fluorescence by sensor-copper particles was noted, and a detection limit at 1 μM was determined, whereas the response time equaled 200 s to total quenching. The nanoprobe was active in the water environment and in the living cells. The selectivity was studied as well, and it was noticed that the response for copper ions is not affected in the presence of $Li^+$, $Na^+$, $Mg^{2+}$, $K^+$, $Ca^{2+}$, $Pb^{2+}$, $Cu^{2+}$, $Zn^{2+}$, $Ag^+$, $Cd^{2+}$, and $Hg^{2+}$ [64].

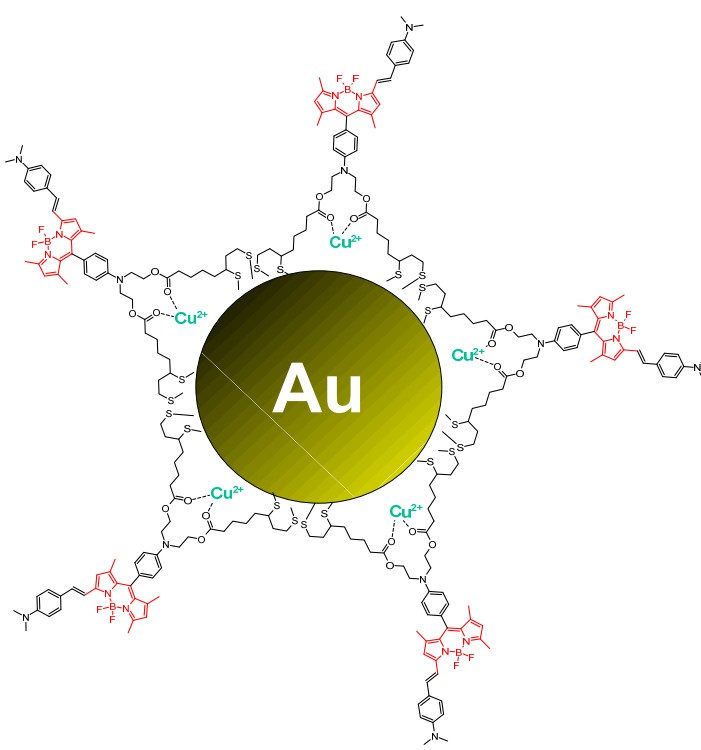

**Figure 10.** The chemical structure of gold nanoparticle surface functionalized with BODIPY dyes as a copper sensor [64].

A nanoprobe for $Hg^{2+}$ ions based on fluorescence quenching was proposed by Son and co-workers [65]. They deposited a highly fluorescent BODIPY dye onto $Au-Fe_3O_4$ nanoparticles (Figure 11). They found that in the presence of mercury ions, nanoprobe emission drops proportionally to the concentration of the analyte. The fluorescence quenching in BODIPYs via coordination with some metal ions such as $Cu^{2+}$, $Hg^{2+}$, or $Cd^{2+}$ was linked with reversed photoinduced electron transfer. The selectivity of the nanoprobe was confirmed through the fluorescence measurements in the presence of $Na^+$, $Mg^{2+}$, $Ca^{2+}$, $Cu^{2+}$, $Fe^{2+}$, $Co^{2+}$, $Cd^{2+}$, and $Ni^{2+}$. Moreover, the authors indicated no pH influence on the emission signal during the determination of the mercury ion level. The detection limit was assigned as 5 ppb [65].

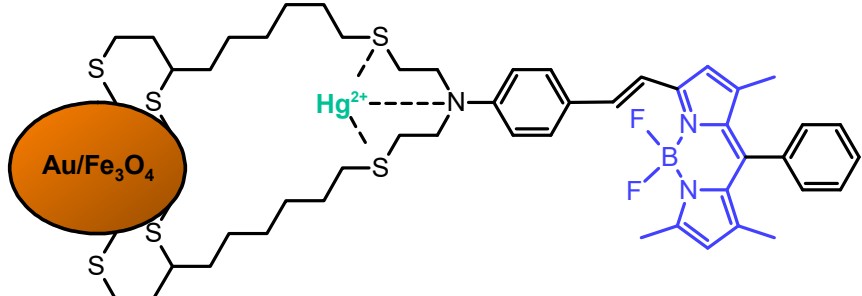

**Figure 11.** The chemical structure of thiolated $Au/Fe_3O_4$ nanocomposite conjugated with BODIPY dye for mercury determination [65].

Another approach was presented by Lo and co-workers, who used a ligand exchange displacement reaction for cysteine detection and determination [66]. The authors built Au nanoparticles with pyridine-BODIPY moieties (Figure 12). The main principle of cysteine detection relies on the nanoparticle ligand exchange due to the high affinity of Au nanoparticles to thiol groups. Weak fluorescent Au-BODIPY nanoparticles ($\Phi$ < 0.01), after exchanging ligands to cysteine, release a fluorescent BODIPY molecule ($\Phi$ = 0.17). The

authors determined a linear response in the range of 0.4–1.4 mM of cysteine, whereas the normal level of cysteine in blood plasma is placed between 240 and 360 μM. The detection limit was assessed as 1.2 μM. Moreover, the selectivity of the developed nanosensor was confirmed in the presence of other amino acids such as Ala, Arg, Asp, Glu, Gly, Hcy, His, Ile, Lys, Met Phe, Pro, Ser, Trp, Tyr, and Val. The authors checked the impact of the nanoprobe on living cells and noted 80% cellular viability after incubation of the nanosensor with RAW264.7 cells within 24 h [66].

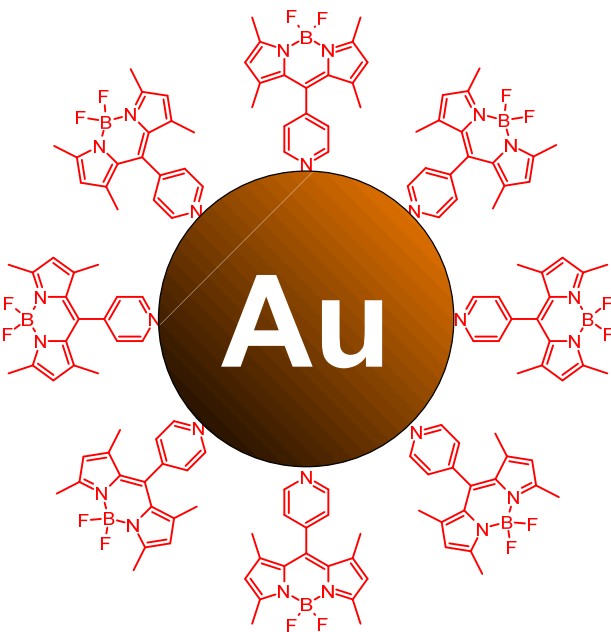

**Figure 12.** The chemical structure of gold nanoparticles functionalized with pyridine-BODIPY moieties [66].

2.1.3. Other Metallic Nanoparticles

Radunz and co-workers developed a ratiometric pH nanosensor based on the fluorescence phenomenon [67]. They employed $NaYF_4$: $Yb^{3+}$, $Tm^{3+}$ nanoparticles with deposited BODIPY dye (Figure 13). This UCN revealed pH-sensitive fluorescence ON/OFF properties depending on $H^+$ concentration. Ratiometric sensors revealed serious advantages. It was observed that, once performed, optimization of calibration gives further reproducible results. The principle of pH measurement with this nanoparticle is based on its excitation of the nanoparticle with 980 nm. Ytterbium ions are excited and transfer their energy to the thulium ion $Tm^{3+}$, which emits light at 451, 475, 646 (R-red), and 805 nm. Light emitted at 475 nm is absorbed by BODIPY dye, which is undergoing excitation and, depending on the pH (protonation of the phenolic component), emits light in the range of 500–600 nm (G-green). The BODIPY fluorescence is turned off at pH values above 9 and turned on during acidification. The G/R ratio gives detailed information on the pH value. The authors deposited the developed nanoparticles onto HydroMed D4 hydrogel, which possesses the capability for water uptake and protons diffusion. The response of the obtained sensor for pH is independent of layer homogeneity and excitation power. The usability of this system was determined according to the measurement of the pH in the biological system. The bacteria *Escherichia coli* was cultured, which acidify their surrounding environment via acid fermentation. Linear dependency was noticed in the sensor working pH range 5–9 [67].

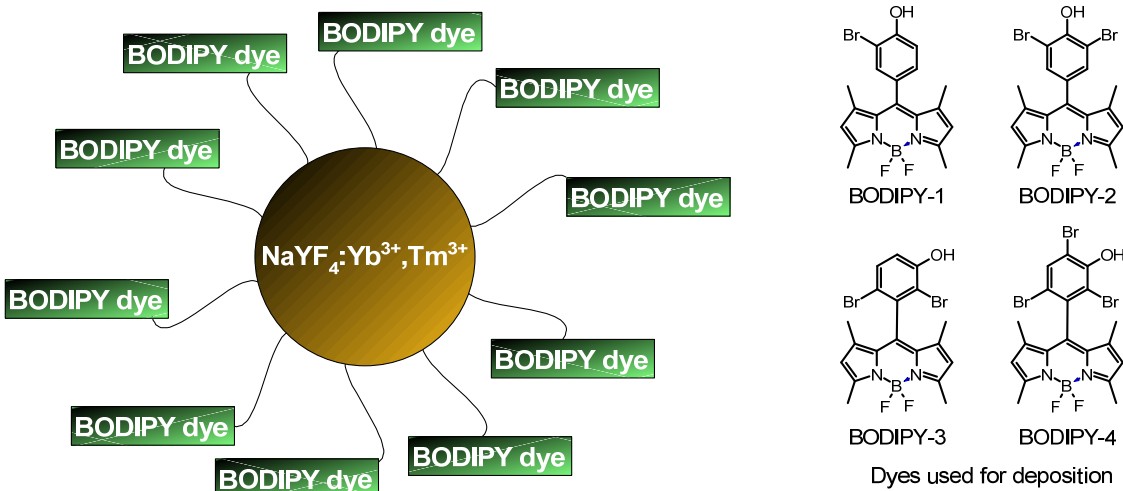

**Figure 13.** The chemical structure of NaYF$_4$: Yb$^{3+}$, Tm$^{3+}$ nanoparticles with deposited BODIPY dye 1–4 [67].

Tong and co-workers developed an arsenate nanosensor based on BODIPY and CeO$_2$ nanowires (Figure 14) [68]. Arsenates cause severe life-threatening effects on the body. Therefore, their concentration in drinking water, especially in spring water, has to be strictly controlled. The World Health Organization regulates the maximal acceptable content of arsenates up to 10 ppb (130 nM). There are a few arsenate quantification methods, but they are expensive, limited by the interferences, time-consuming, etc. The proposed nanoprobe was based on the BODIPY fluorescence and its quenching by the nanowire. BODIPY fluorescence quenching by CeO$_2$ is caused by oxidative electron transfer from the excited dye to the nanowire. The studied nanowire revealed dimensions of 120 ± 15 nm × 8.0 ± 2.5 nm. BODIPY dye was deposited on these surfaces via an ATP linker. The nanosensor needed only 10 min to respond, which is based on the competitive sorption of BODIPY and arsenates. In the presence of arsenates, the BODIPY was released, and the fluorescence signal increased. The linear dependence between the concentration of arsenates and the intensity of BODIPY fluorescence in the ranges of 20–150 and 150–1000 nM was determined at the detection limit estimated to be 7 nM. Moreover, the authors assessed the stability of the response in the pH range of 3–7 as well as selectivity for arsenates in the presence of arsenate or phosphate ions. Recognition of arsenates and phosphates was provided by an ATP linker with optimal phosphate chain length [68].

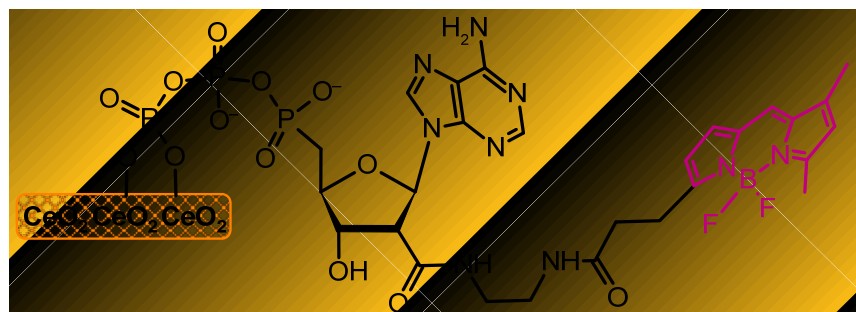

**Figure 14.** The chemical structure of cerium(IV) oxide nanowires conjugated with BODIPY dye by ATP linker [68].

To sum up:

- Different strategies were used for conjugation of BODIPY dyes to metallic nanoparticles through direct coordination or through indirect attachment via silica or thiolate linkers.
- BODIPY/metallic nanoparticles sensors can act as fluorescent probes.
- BODIPY dyes in conjugation with gold and magnetic nanoparticles were mostly used to determine heavy metal ions in aqueous solutions.

- Upconverted nanoparticles surface functionalized with BODIPY dyes can be potentially used as pH sensors.

### 2.2. Cancer Diagnosis and Therapy

### 2.2.1. Gold Nanoparticles

Adarsh and co-workers obtained a nanoprobe for cancer cells based on surface-enhanced Raman scattering (SERS) [69]. It was based on synthesized amino aza-BODIPY deposited on a gold nanoparticle (Figure 15). The obtained conjugate revealed an intense Raman signal with a detection rate at the concentration of 0.4 μM. To increase the selectivity, the Raman nanoprobe was functionalized with EGFR (epidermal growth factor receptor) and closed in PEG. Performed studies indicated high recognition of human fibrosarcoma cancer cells (HT1080 line), whereas simultaneously normal human cells (3T3L1) remained unaffected [69].

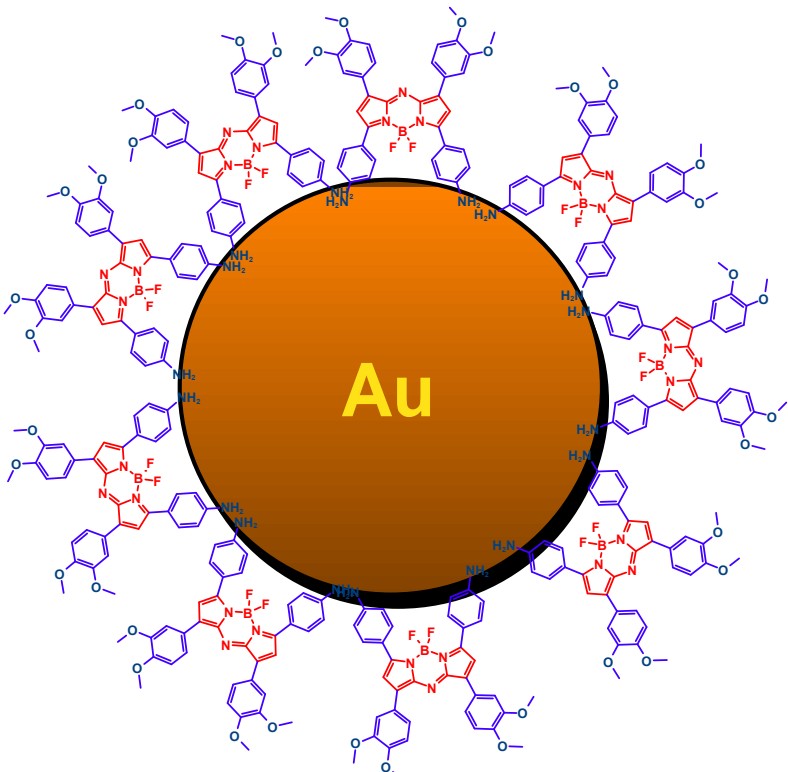

**Figure 15.** The chemical structure of human fibrosarcoma cancer cells nanoprobe based on aromatic-substituted aza-BODIPY dyes and gold nanoparticle [69].

Kim and co-workers developed a drug delivery system based on gold monolayer nanoparticles [70]. The authors used "hydrophobic pockets" reported previously by Lucarini and Pasquato [71]. On the gold surface, zwitterionic chains were attached, and in the formed pockets, BODIPY molecules were loaded as a fluorescent probe (Figure 16). The authors concluded that dye is delivered by the monolayer-membrane transfer process. Moreover, only low uptake of the nanocarrier bearing zwitterionic chains into the MCF-7 cells (human breast cancer) was confirmed. Efficient delivery into the cytosol was noticed after 2 h of particles incubation with the cells. BODIPY-loaded nanoparticles sustained stability for over a month [70].

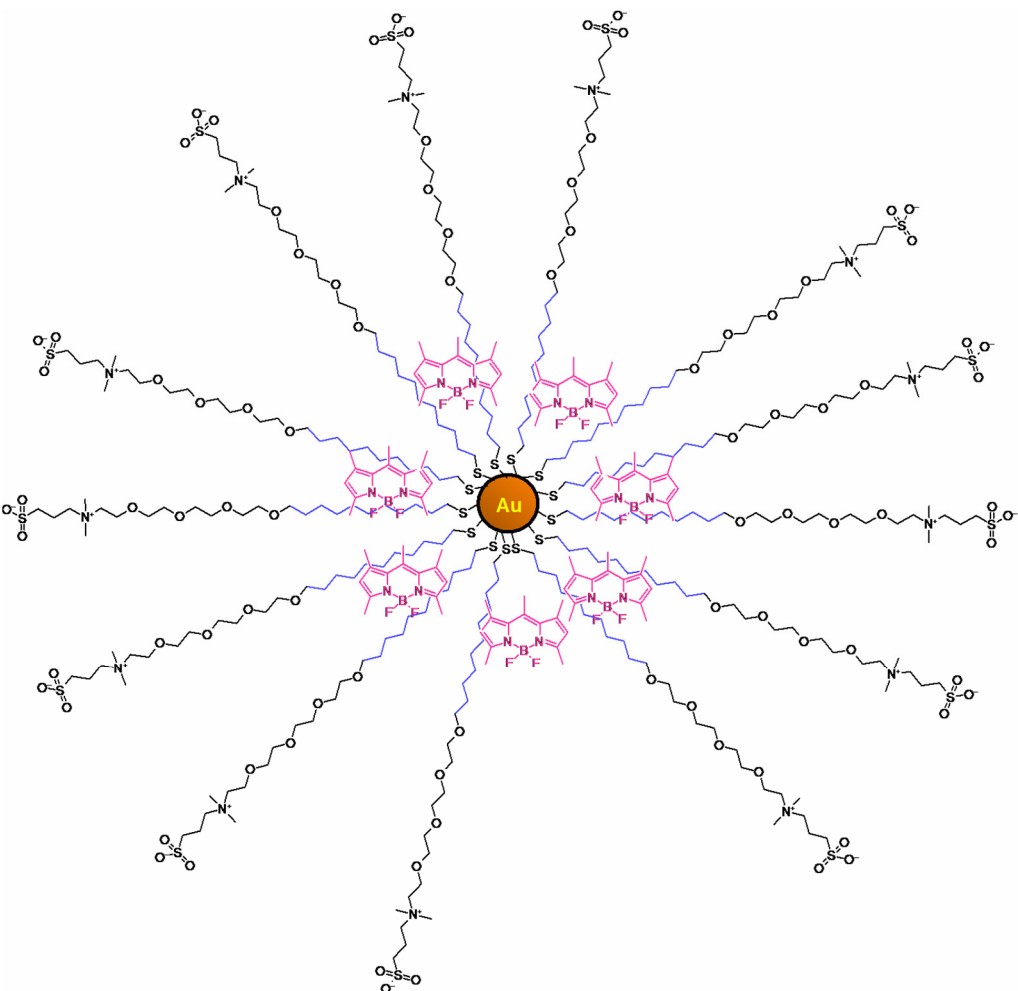

**Figure 16.** The chemical structure of gold nanoparticle with zwitterionic chains forming pockets where BODIPY molecules were loaded [70].

Kumar and co-workers studied the usability of new gold nanoparticles coated with BODIPY and L-tryptophan in photodynamic therapy (Figure 17) [72]. They concluded that the new nanoparticle is an efficient singlet oxygen generator with a quantum yield ΦΔ of 0.46. Moreover, in dark conditions, the nanoparticle was not active against C6 rat glioma cells (80% viability), whereas the viability decreased to ca. 25% after excitation of the nanoagent at a concentration of 100 μg/mL with light. The influence of the nanoagent on normal living cells was checked via incubation with normal fibroblasts derived from mice (L292). As a result, the viability of cells was found to be greater than 80%, pointing to high biocompatibility [72].

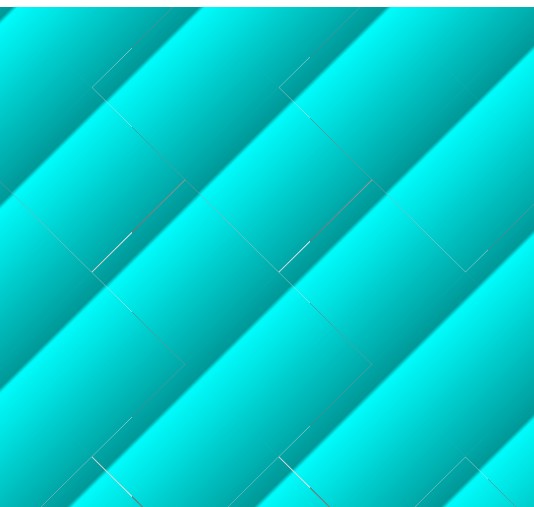

**Figure 17.** The chemical structure of gold nanoparticle coated with BODIPY dyes and L-tryptophan [72].

### 2.2.2. Magnetic Nanoparticles

Kainz and co-workers proposed nanomagnets for biomedical applications [73]. On the magnetic carbon coated with cobalt nanoparticles, authors constructed multifunctional particles bearing fluorescent BODIPY moieties and dendrimeric components (Figure 18). BODIPY molecules are known as very efficient light emitters, and the dendrimeric component provides good solubility. Despite the literature reporting quenching of fluorescence dyes deposited on the carbon surfaces, the authors noted the same level of fluorescence for bounded and free BODIPY dyes, making the nanoparticles suitable for MRI and fluorescence diagnosis. Non-covalently bonded drugs (BODIPY) can be triggered in a controlled manner via temperature and changing of the solvent or pH, allowing the presented nanomagnets to be used for theranostic purposes [73].

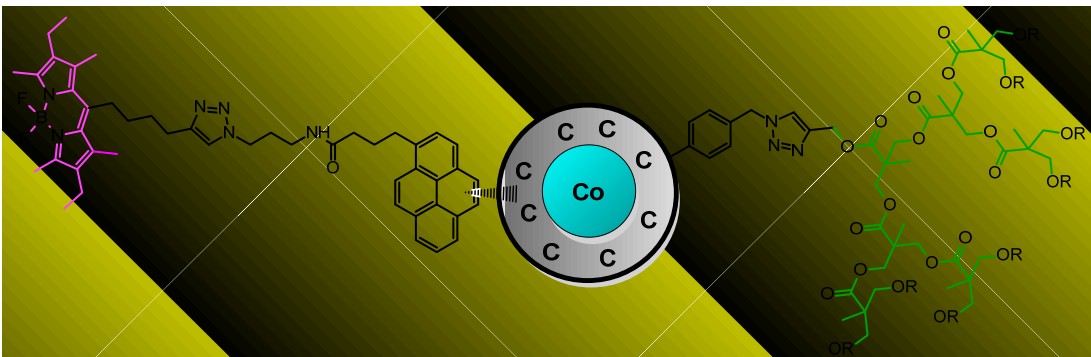

**Figure 18.** The chemical structure of magnetic carbon coated with cobalt nanoparticles conjugated with BODIPY dye and dendrimeric moieties [73].

An interesting utility of BODIPY immobilized onto iron oxide superparamagnetic nanoparticles (SPIONs) with chelated $Ni^{2+}$ ions was reported by Maltas and co-workers (Figure 19) [74]. They adjusted the formation of an immobilized metal affinity (IMA) adsorbent for protein purification via the adsorption process in the chromatography method. To assemble this IMA absorbent, 10 mg of SPIONs were decorated with 29.7 µM of previously prepared BODIPY (8-(2,6-diethyl-1,3,5,7-tetramethyl-4-bora-3a,4a-diaza-s-indacene) benzoylchloride) and then 738 µmol of Ni(II) ions from nickel nitrate were adsorbed by 10 mg of SPIONs/BODIPY particles with an adsorption rate of 74.3%. Cytochrome c was used as a model for binding characterization due to the formulation of fluorescent SPIONs/BODIPY/Ni-Cyt c agglomerates. The authors concluded that the complexation of nickel ions by nanoparticles increases the binding ability, which was evaluated as 170 µg at pH 7.4 per 10 mg of SPIONs/BODIPY/Ni complex [74].

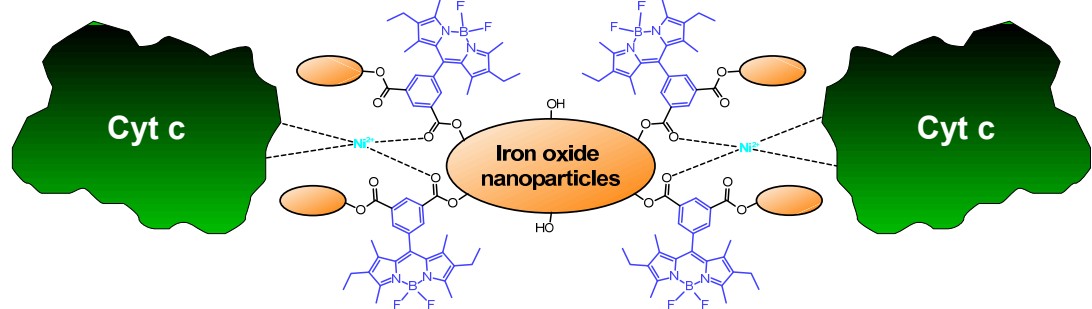

**Figure 19.** The chemical structure of SPIONs conjugates with cytochrome c [74].

### 2.2.3. Zirconium and Metal-Organic Framework Nanoparticles

Oh and co-workers built a PDT usable nanoagent based on zirconium hydroxide nanoparticles bearing carboxyphenylporphyrin-PCN-222 and a BODIPY dye (Figure 20) [75]. Zirconium forms a MOF thanks to the attached BODIPY and improves singlet oxygen production in comparison to pure PCN-222. BODIPY was conjugated via the "SALI" (solvent-assisted ligand incorporation) process. The obtained nanoagent excited with white LED light revealed an $IC_{50}$ value of ca. 5 nM against human breast adenocarcinoma cells (MCF-7) and 9 nM against mouse melanoma (B16F10). The improvement of the activity in the light phase in contrast to the dark one was 2000 times for MCF-7 and 10,000 times for B16F10. Moreover, the authors observed that nanoparticles enter the cancer cells and spread in the cytosol [75].

MOF-BODIPY nanoparticles (UiO-PDT) based on zirconium were obtained by Wang and co-workers via solvent-assisted ligands exchange [76]. The MOFs were based on zirconium (inorganic part) and benzenedicarboxylate (organic part). Their structure is presented in Figure 21. However, the BODIPY molecule was modified with iodine substituents to provide high efficacy of singlet oxygen formation. TEM (transmission electron microscopy) and SEM (scanning electron microscopy) analyses indicated that the MOF reveals an octahedral shape. The incorporation of BODIPY in the MOF resulted in a slight shift of the absorption band and a significant red shift of the emission (from 549 to 563 nm). The fluorescence ability was evaluated after incubation of UiO-PDT with mouse melanoma cells (B16F10). The authors noticed red emission of the UiO-PDT and free BODIPY placed in the cytoplasm, indicating that particles entered the cells. They also examined biocompatibility as an important factor of every potential drug. MOFs modified with BODIPY expressed low cytotoxicity against several cell lines (B16F10, C26, CT26) at a concentration of 1 mg/mL. In addition, the photocytotoxicity of obtained MOFs towards the above-mentioned cell lines was also evaluated. Simultaneously, the authors performed control experiments with a bare MOF and bare BODIPY. High photoactivity of bare BODIPY was observed ($IC_{50}$ < 0.8 μg/mL), while in dark conditions, the studied PS remained inactive. The control study with a bare MOF indicated no significant activity both under irradiation and in the dark. MOFs modified with BODIPY revealed high photoactivity against the above-mentioned cell lines with the $IC_{50}$ values within the range of 0.51–1.15 μg/mL [76]. Zhang and co-workers continued studies with UiO-PDT nanoparticles and used them as computed tomography (CT) imaging agents. In CT imaging, it is crucial to obtain the desired contrast deriving from molecules/atoms with high X-ray attenuation, i.e., iodine, barium, and bismuth. The authors proved the lack of cytotoxicity in an in vitro experiment with HepG2 cells. Next, they performed detailed in vivo experiments with a rat model. The hematoxylin and eosin staining of vital organs (heart, liver, spleen, lung, and kidney) was performed. Function markers of liver, kidney, gallbladder indicator and some blood parameters (hemolysis, prothrombin time, and activated partial thromboplastin time) were measured. Based on the performed experiments, the authors concluded that UiO-PDT nanoparticles reveal high biocompatibility up to a dose of 100 mg/kg b.w. The contrasting ability of nanoparticles was evaluated via visualization of tumor in rats bearing hepatomas.

The highest signal was observed 24 h after intravenous agent injection. The hepatic tumor was indicated with a brighter signal when compared to the surrounding tissues, although not as bright as bones [77].

**Figure 20.** A fragment of the chemical structure of zirconium hydroxide-5,10,15,20-tetra(carboxyphenyl) porphyrin metal-organic framework-PCN-222, decorated with BODIPY dyes [75]. Explicit hydrogens of the $[Zr_6(\mu_3\text{-OH})_8(\text{-OH})_8]^{8+}$ cluster are not shown in the structure for clarity.

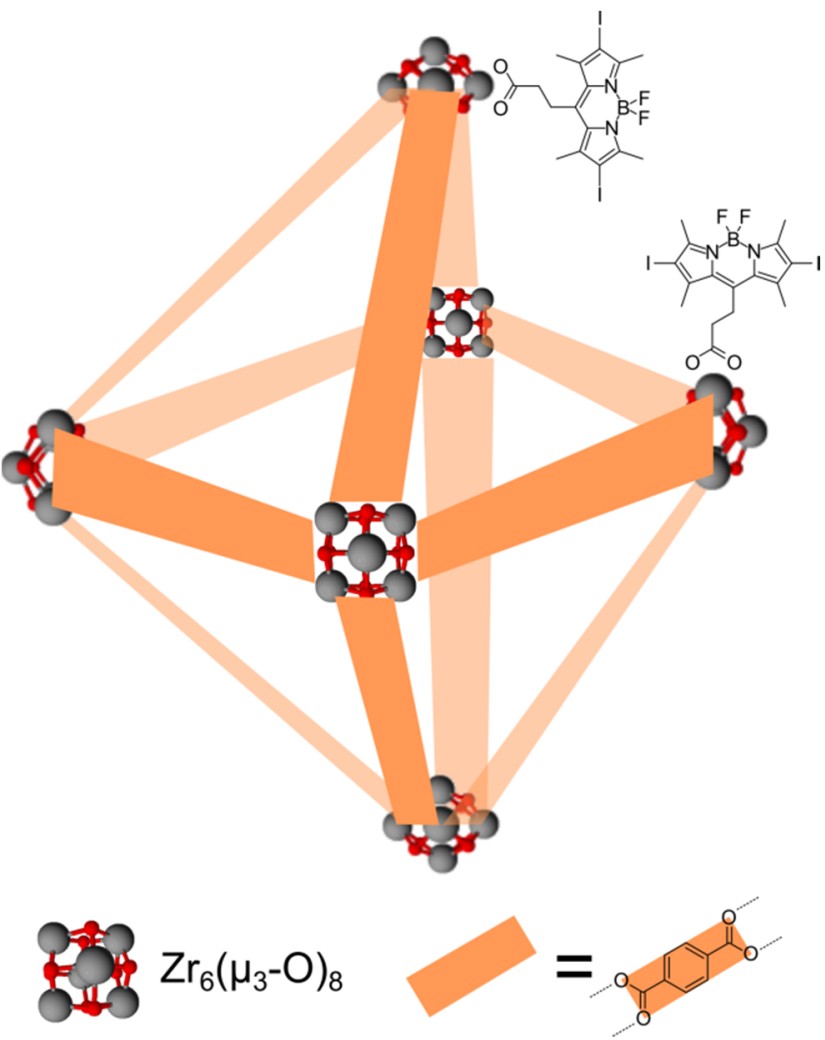

**Figure 21.** The simplified chemical structure of UiO-66 MOF decorated with BODIPY dyes [76].

　　　MOFs were adopted to deliver an imaging and therapeutic agent to cancer cells. Within the MOF class, Taylor-Pashow and co-workers chose the MIL representatives (Materials of Institute Lavoisier) formed via trivalent metal linked with carboxylate bridges (firstly described by Férey and co-workers [78,79]). The authors used MIL based on $Fe^{3+}$ ions with the formula $Fe_3$-$(\mu_3$-O)Cl$(H_2O)_2$(BDC)$_3$, where BDC was terephthalic acid (Figure 22). The obtained nano-carrier revealed porous octahedron morphology with a diameter ca. 200 nm and surface area within the range 3700–4535 $m^2$/g. The particles were modified with 2-aminoterephthalic acid (17.5 mol %), which provides an anchor for a BODIPY dye (loaded with efficiency 20.9–40.3%). The BODIPY molecule deposited onto MIL did not reveal fluorescence due to quenching by the d-d conversion of iron ions, but the released molecule strongly emits light. Due to the short release time of BODIPY ($t_{0.5}$ = 2.5 h) and ethoxysuccinato-cisplatin loaded on MIL ($t_{0.5}$ = 1.5 h) in PBS (phosphate-buffered saline), the authors coated loaded nanoparticles with a silica layer. This procedure enhanced the release time to parameters suitable for medical applications—ca. 14 h for BODIPY and ca. 16 h for cisplatin. The authors performed in vitro studies with the human colon adenocarcinoma cell line (HT-29). They concluded that BODIPY derivatives were efficiently delivered into the cells in which they can be visualized, whereas the free dyes did not possess this feature. Simultaneously, it was concluded that cisplatin loaded on the particles showed activity similar to free cisplatin [79].

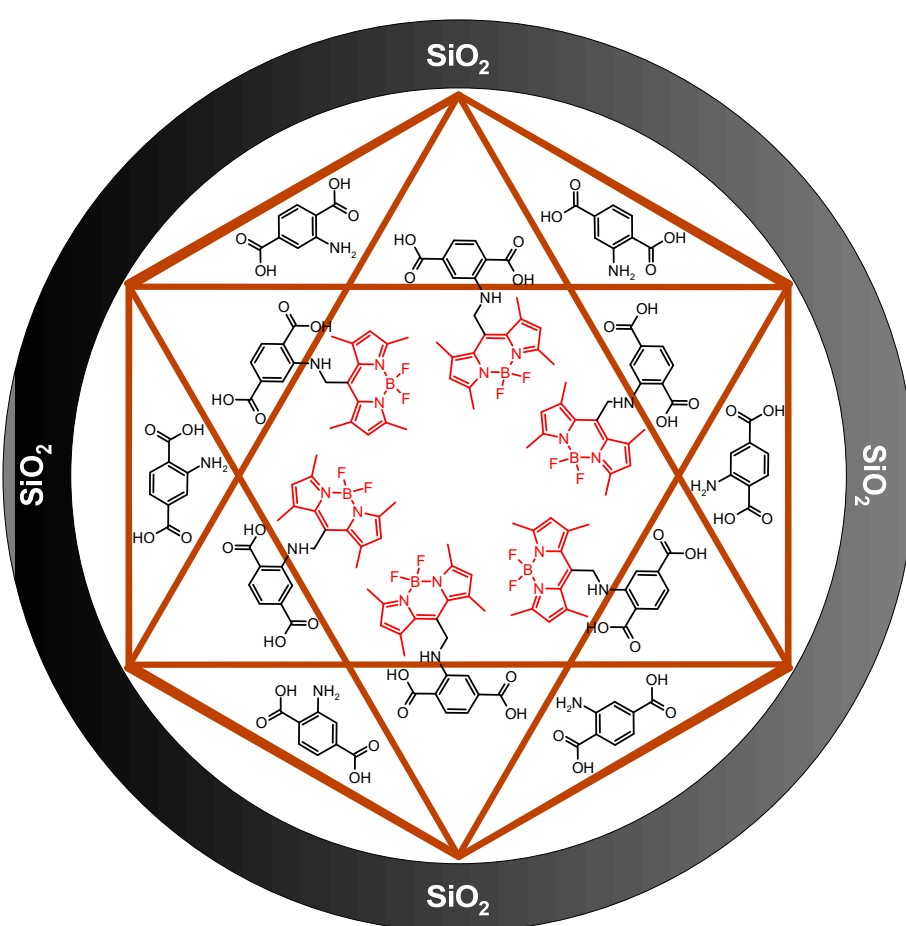

**Figure 22.** The representative chemical structure of Materials Institute Lavoisier (MILs). MIL-101 (Fe) was obtained with the use of 2-aminoterephthalic acid and modified with BODIPY dyes coated with silicon(IV) dioxide [79]. The orange lines represent the 2-aminoterephthalic acid moieties, and the six nodes where they meet are occupied by $Fe_3(\mu_3\text{-O})Cl(H_2O)_2$ clusters.

### 2.2.4. Other Metallic Nanoparticles

Metal nanoparticles bearing a BODIPY photosensitizer were employed to overcome PDT limitations related to low light penetration of tissues. The delivery of the light is essential to excite the PS and produce singlet oxygen—the anticancer agent. Thus, Ozdemir and co-workers applied the persistent fluorescence phenomenon via a FRET-matching BODIPY dye [80]. They built zinc-germanogallate-based ($Zn_{2.78}Ga_{1.68}Ge_{1.00}O_8{:}Cr_{0.01},Pr_{0.02}$) BODIPY-functionalized nanoparticles (Figure 23). The obtained nanoagent was excited with light at 254 nm before being introduced to the tumor. Persistent luminescence is caused by a cascade of electron recombination in the nanoparticle. Firstly, after photons absorption, excited electrons are captured by native defects of zinc-germanogallate. When the excitation process is finished, the electrons leave holes. Due to the spectral overlap of $Cr^{3+}$ ions, the released energy is transferred to these ions in a non-radiative manner. Electrons located at 3d orbitals of chromium ions are excited to triplet states and move to deeper traps that, after persistent energy transfer, stop releasing energy and are able to emit light persistently. Emitted light (at ca. 500 nm) is suitable to be absorbed by the attached BODIPY dye, which then generates singlet oxygen up to 80 min after excitation. In vivo experiments concluded a 15% reduction of the tumor volume. This modest pilotage result might open up a new pathway for PDT development [80,81].

**Figure 23.** The chemical structure of zinc-germanogallate-based BODIPY-functionalized nanoparticles [80].

Silver-based BODIPY-bearing nanoparticles were developed for selective bacteria recognizing and combating (Figure 24). Zhang and co-workers linked iodinated BODIPY via galactose-polymer with silver nanoparticles [82]. Iodine atoms enriched singlet oxygen production via a phenomenon known as the "heavy atom effect". Galactose-polymer was responsible for selective binding with bacteria cells. The platform was formed with silver nanoparticles revealing some antibacterial effects. The authors confirmed, with a laser scanning confocal microscope, efficient gathering of nanoparticles in the bacteria in comparison to mammalian cells. The bactericidal activity of obtained nanoparticles activated with light was assigned to MIC = 50 pmol/mL, whereas the MIC value of these particles in dark conditions was 2500 pmol/mL. The authors treated the NIH/3T3 cell line with obtained nanoparticles (400 pmol/mL) and noticed that over 80% of the cells survived. BODIPY-Ag nanoparticles (10 nmol/mL) incubated with erythrocytes caused negligible hemolysis. Therefore, it was concluded that the nanoparticles revealed high biocompatibility [82].

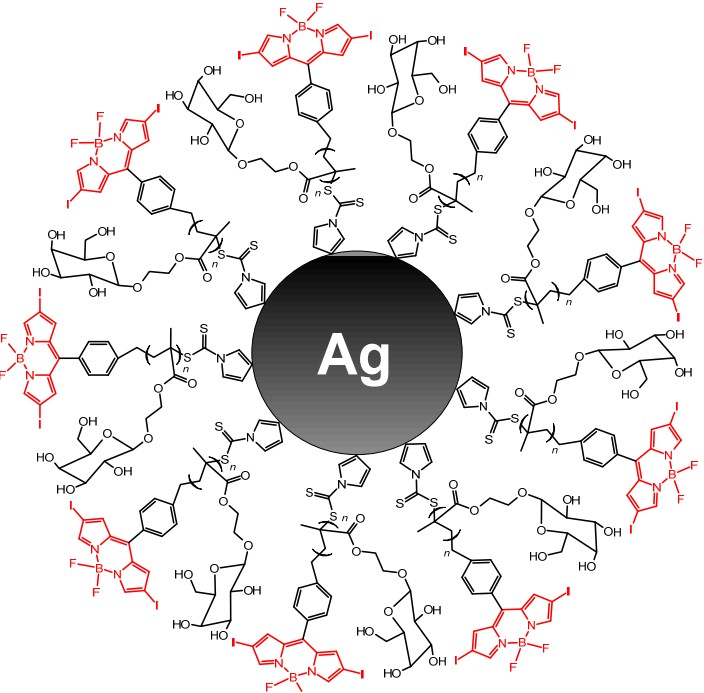

**Figure 24.** The chemical structure of silver nanoparticles functionalized with iodinated BODIPY linked to galactose-polymer [82].

In search of new tools for PTT and PDT implementation, Liu and co-workers developed an aza-BODIPY probe containing mesoporous black anatase titanium dioxide

nanoparticles named MTAB (Figure 25) [83]. The synthesis of MTAB occurred under analytical surveillance using such methods as XRD, [1]H NMR, and Fourier-transform infrared (FTIR) spectroscopy. First, the authors obtained mesoporous anatase $TiO_2$ nanoparticles that consequently underwent aluminum reduction to produce mesoporous black anatase nanoparticles (MT). The latter ones were decorated with assembled heretofore aza-BODIPY (AB) structures, resulting in 1.38 wt% loaded nanoplatforms (confirmed via thermal gravimetric analysis) used to arrange 5 mg/mL suspension for further experiments. SEM and TEM studies unveiled that these heterogeneous nanoplatforms constitute nanospheres rich in mesopores with an approximate size of 100 nm (the hydrodynamic size was 155 nm if dispersed in deionized water). At the same time, the Brunauer–Emmett–Teller (BET) method was availed of to determine their surface area and defined it as 126.5 $m^2/g$. Additionally, the UV-vis absorption spectrum showed two distinctive peaks at 389 and 820 nm. Photothermal and photodynamic qualities were measured under single-wavelength NIR light laser irradiation. As a result, MTAB irradiated with 808 nm laser light (0.5 $W/cm^2$, 10 min) caused the most eminent effectiveness in terms of tumor cell extermination according to cytotoxicity outcomes, demonstrating 33.98% photothermal conversion efficiency and significant increase of ROS generation (observed via confocal fluorescence microscopy). Continuous (14 days) observation of mice inflicted with MTAB-mediated phototherapy led to the conclusion of a 94.8% tumor eradication rate, while the mice deprived of NIR irradiation suffered from cancer progression: without photocatalytic stimuli, MTAB did not provide any impact on the tumor growth [83].

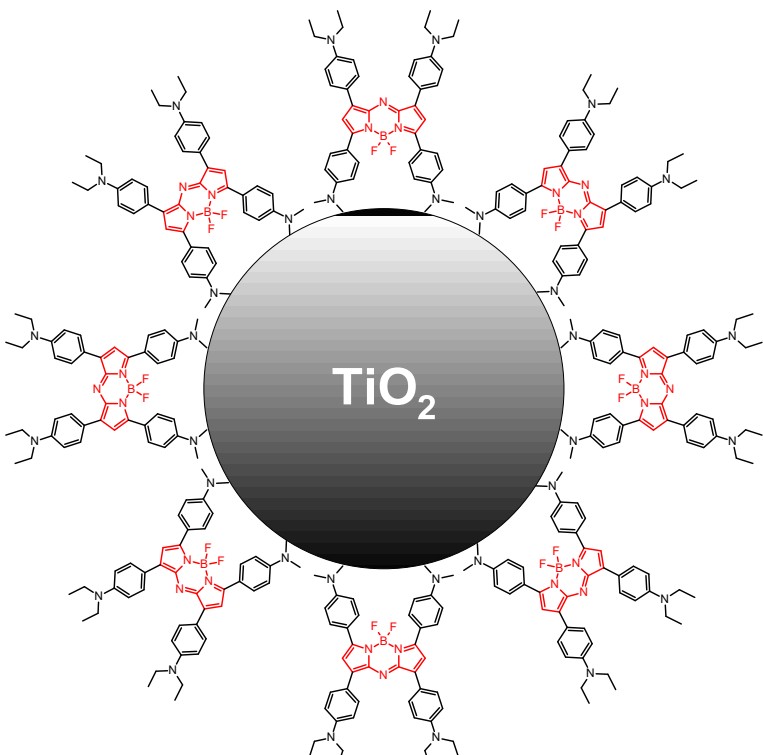

**Figure 25.** Schematic representation of the chemical structure of "black" titanium(IV) oxide nanoparticle functionalized with aza-BODIPY dyes through adsorption of the dye on the nanoparticle surface [83].

To sum up:

- Gold nanoparticles/BODIPY conjugates were applied in in vitro studies as Raman or fluorescent sensors or co-photosensitizers.
- Fluorescent probes were obtained by combining magnetic nanoparticles and BODIPY dyes.
- Zirconium-based metal-organic frameworks functionalized with BODIPY dyes were used as photosensitizers in in vitro photocytotoxicity studies and revealed high photoactivity.

- A high antibacterial effect was achieved with the use of Ag nanoparticles/BODIPY dyes in photodynamic inactivation in vitro tests.

### 2.3. The Relationship of the Chemical Structure on Physicochemical Properties and Activity of BODIPY

Taking into consideration the results of studies reviewed in this paper, one can conclude some relationships and impacts of the chemical structure of BODIPY derivatives on their medical use as well as physicochemical properties, including fluorescence and binding abilities. The uniqueness of BODIPY is the result of its chemical structure, where a connection of two pyrrole rings with a methylene bridge, also called the "*meso*" position is present. In addition, two pyrrole nitrogen atoms can also be linked with a difluoroboron bridge that forms the third ring. Moreover, BODIPY dyes are often symmetrically functionalized with various functional groups within the pyrrole rings or at the *meso* position to achieve the desired properties.

The fluorescence emission ability of BODIPY dyes is analyzed in sensing studies. BODIPY derivatives substituted with the phenyl rings in *meso* position are most often used as fluorophores. The replacement of phenyl with the pentafluorophenyl group significantly increases the fluorescence [59]. The effect is comparable when the BODIPY pyrrole rings are substituted in positions 3 or 5 with styryl groups [64,65]. However, the amide linkers attached to the pyrrole BODIPY rings can sometimes hamper the fluorescence through the PET phenomenon [60]. BODIPY's peripheral substitution also brings other benefits. In various studies, the BODIPY dyes have been attached to the coated or uncoated metallic nanoparticles by phenyl groups in their *meso* position [60,62] or by styryl/alkyl linkers at pyrrole rings [61,65,73]. It was also demonstrated that the heteroaromatic pyridyl substituent in the BODIPY periphery allows the dye to be attached directly to the nanoparticle surface [66]. What is more, the phenyl group substituted in *meso* position can be protonated, which also changes the fluorescence properties under the influence of different pH [67].

BODIPY dyes with bulky peripheral substituents have been applied in cancer diagnostics [69]. It was possible when the 3,5-dimethyl groups at pyrrole rings were substituted with aromatic ones. It was found that such a change in the chemical structure improves the solubility in common organic solvents used in the preparation of BODIPY/metallic nanoparticle hybrid nanosystems and allows the direct attachment of dyes to the surface of gold nanoparticles. In cancer diagnostics, the fluorophore moiety consists mainly of *meso*-methyl derivatives of dipyrromethene instead of phenyl-substituted ones [70,72]. Substitution of the 4th position of the BODIPY pyrrole rings with the halogen atom (mostly iodide) results in an increase of singlet oxygen generation after irradiation of the system. The phenomenon explaining such an effect is commonly called the "heavy atom effect". This kind of modification is of great importance for further applications of BODIPY dyes in photodynamic therapy against cancer cells or bacteria strains. The iodide-substituted derivatives reveal no fluorescence ability but instead high efficiency of singlet oxygen production when irradiated with light at proper wavelengths. In addition, the absorbance of dyes is red-shifted in comparison to unsubstituted derivatives. BODIPY dyes representing such structures have been researched in various in vitro cytotoxicity studies against, i.e., human breast adenocarcinoma cells, mouse melanoma, human colon adenocarcinoma, and against *P. aeruginosa* and *S. aureus* [75,76,80–82].

A schematic representation of the effects of individual BODIPY structural elements on their physicochemical and spectral properties is shown in Figure 26.

**Figure 26.** The influence of the chemical structure elements of BODIPY dyes on their properties.

## 3. Conclusions

Recent scientific results in the field of novel materials based on metal-based nanostructures and BODIPY derivatives were gathered in this review. Multiple research documents were analyzed concerning the latest discoveries within the scope of BODIPY-based nanomaterials with particular emphasis on their utilization for diagnostic sensing as well as cancer diagnostics and therapy. The collected literature data on the mentioned materials were presented in order to draw the attention of the scientific community to their practical applications, consistently elucidate the topic, and inspire fellow researchers for new findings concerning their expediency in the diagnosis and treatment of tumor ailments as well as in sensing of heavy metals.

Many of the discussed BODIPY-modified nanoparticles have found applications in determining biologically significant ions. One interesting example concerns pH detection and toxic chromium(VI) ions detection and their elimination from aqueous solutions. Another example constitutes superparamagnetic/BODIPY connections, which can be applied in the complexation of nickel ions, giving access to unique adsorbents for protein purification in the immobilized metal affinity chromatography method. Other nanoparticles carrying BODIPYs manifested highly selective and sensitive photoreaction with cysteine molecules in biological conditions or constituted a reliable tool for determining the immunosuppressive tacrolimus drug in narrow concentration ranges.

Particular attention was given to cancer diagnosis and treatment techniques where metallic nanoparticles/BODIPY conjunctions firmly distinguished themselves. The aforementioned examples of MOFs with BODIPY were successfully tested for many medical applications. Among these applications were tumor visualization, promoting dye internalization by cancer cells, photothermal imaging, computed and photoacoustic tomography, and significant photodynamic tumor reduction due to improved light-harvesting properties and the heavy-atom effect. The experiments on carbon-coated cobalt nanomagnets bearing noncovalently joined fluorescent BODIPY moieties and covalently attached dendrimeric components showed their expediency as transporting units for multifunctionalization with drug and targeting molecules, especially in cancer treatment. Recently discovered aza-BODIPY probe-containing mesoporous black anatase titanium dioxide nanoplatforms represent highly potential theranostic instruments for effective photothermal and photodynamic cancer therapy. Attachment of another aza-BODIPY derivative to gold nanoparticles allowed ultrasensitive recognition of human fibrosarcoma cell lines to be achieved. Other functionalized golden nanocarriers with BODIPY as guest compounds entrapped in hydrophobic pockets demonstrated promising capabilities for targeting drug delivery in tumor tissues. Silver nanoparticles with iodinated BODIPY molecules inserted through galactose polymer revealed high efficacy in eradicating bacterial infection, bypassing even established drug resistance. In Table 1 we summarize the data collected in this review.

**Table 1.** A tabular summary of the data described in this review.

| BODIPY Core | Metallic Nanoparticle | Type of Connection | Application | Ref. |
|---|---|---|---|---|
| Pentafluorophenyl-substituted BODIPY Styryl-substituted BODIPY | Ferrite/polystyrene | Physical deposition/adsorption | Determination of tacrolimus | [59] |
| *Meso*-phenyl-substituted BODIPY | Ni/silica | Covalent attachment via amide bonding | $Pb^{2+}$ sensing | [60] |
| *Meso*-phenyl- and styryl-substituted BODIPY | $Fe_3O_4$/silica | Covalent attachment via amide bonding | $Pb^{2+}$ sensing | [61] |
| *Meso*-phenyl-substituted BODIPY | $Fe_3O_4$/silica | Covalent attachment via amine bonding | $Cr^{6+}$ sensing | [62] |
| *Meso*-phenyl- and styryl-substituted BODIPY | Au nanoparticles | Covalent attachment via ester bonding | $Cu^{2+}$ sensing | [64] |
| *Meso*-phenyl- and styryl-substituted BODIPY | Au nanoparticles | Covalent attachment via thienyl bonding | $Hg^{2+}$ sensing | [65] |
| *Meso*-pyridyl-substituted BODIPY | Au nanoparticles | Electrostatic interactions | Aminoacids sensing (cysteine) | [66] |
| *Meso*-bromohydroxyphenyl-substituted BODIPY | $NaYF_4$:$Yb^{3+}$, $Tm^{3+}$ nanoparticles | Physical deposition/adsorption | ratiometric pH-nanosensor | [67] |
| Unsubstituted BODIPY | $CeO_2$ nanowires | Covalent attachment via amide bonding | Arsenates sensing | [68] |
| Aromatic amino aza-BODIPY | Au nanoparticles | Electrostatic interactions | Nanoprobe for cancer cells | [69] |
| *Meso*-methyl-substituted BODIPY | Au nanoparticles | Physical deposition/adsorption | Cancer diagnosis | [70] |
| *Meso*-methyl-substituted BODIPY | Au nanoparticles | Physical deposition/adsorption | Photodynamic therapy | [72] |
| Ethyl-substituted BODIPY | Carbon-coated Co nanoparticles | Covalent attachment via alkyne-azide conjugation (triazole) | Cancer diagnosis | [73] |
| *Meso*-phenyl- and ethyl-substituted BODIPY | Iron oxide nanoparticles (SPIONs) | Covalent attachment via ester bonding | $Ni^{2+}$ complexation | [74] |
| *Meso*-phenyl- and iodide-substituted BODIPY | Zirconium hydroxide nanoparticles (MOFs) | Covalent attachment via ester bonding | Photodynamic therapy | [75] |
| *Meso*-alkyl- and iodide-substituted BODIPY | Zirconium hydroxide nanoparticles (MOFs) | Covalent attachment via ester bonding | Photodynamic therapy | [76,77] |
| *Meso*-ethyl-substituted BODIPY | Silica-coated Fe-based MOFs | Physical deposition/adsorption | Photodynamic therapy | [79] |
| *Meso*-phenyl-, iodide-, and styryl-substituted BODIPY | Zinc-germanogallate-based nanoparticles | Covalent attachment via ether bonding | Photodynamic therapy | [80,81] |
| *Meso*-phenyl- and iodide-substituted BODIPY | Silver nanoparticles | Covalent attachment via ester bonding | Photodynamic inactivation of bacteria | [82] |
| Aromatic aza-BODIPY | Titanium oxide nanoparticles | Physical deposition/adsorption | Photodynamic therapy | [83] |

The collected data encourage further explorations in the area of metallic nanoparticles/BODIPY conjugates since the provided examples proved their practicality and opened new horizons for discoveries of even more potent biologically active photosensitive nanostructures.

**Funding:** The authors gratefully acknowledge National Science Centre Poland for financial support (Grant no. 2016/21/B/NZ9/00783).

**Institutional Review Board Statement:** Not applicable.

**Informed Consent Statement:** Not applicable.

**Data Availability Statement:** Not applicable.

**Acknowledgments:** Stepan Sysak is a participant of the STER Internationalisation of Doctoral Schools Programme from NAWA Polish National Agency for Academic Exchange No. PPI/STE/2020/1/00014/DEC/02.

**Conflicts of Interest:** The authors declare no conflict of interest.

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
