# Peer review of "BODIPY-Based Nanomaterials—Sensing and Biomedical Applications"

_applsci, doi:10.3390/app12157815_

Round 1

Reviewer 1 Report

The authors present in this manuscript a comprehensive review of nanomaterials for biomedical applications based on BODIPY fluorophores. The article is well-written, concise, and provides a good perspective on the research topic.

I consider this contribution also timely since it deals with a topic of current interest including very recent contributions in the field.

Typos:

In Fig. 15. aza-BODIPY rather than BODIPY .... gold nanoparticles.

Reviewer 2 Report

This paper reviewed recent publications connected to BODIPY-conjugated metallic nanoparticles. It firstly gives an introduction to nanotechnology and nanomaterials in medicine as well as metallic nanoparticles in medicine with clear figures for classification. And then they give a quite detailed characterization of selected nanoparticles, nanoparticles involved in photodynamic therapy and BODIPY dyes. Each part is well organized by starting from a basic and detailed introduction and ending with concise summary statements. The whole introduction part is clear and lucid with the points well highlighted, which is quite instructive.

The main text mainly reviewed metallic nanoparticles@BODIPY from the sensing applications and the cancer diagnostic and therapy. For the sensing applications, they mostly reviewed the nanoparticle BODIPY conjugates based on magnetic, gold and some other metallic nanoparticles like Tm3+ nanoparticles and CeO2 nanowires. For cancer diagnosis and therapy, gold, magnetic, zirconia and metal-organic framework nanoparticles are well summarized. Other examples are also mentioned including metallic nanoparticles like silver-based nanoparticles, zinc-germanogallate-based nanoparticles and so on. Each part is presented with plenty of literatures information, refined description and natty schematic figures. Also, the key information in both areas are succinctly summarized item by item.

This is an excellent review article with sufficient and comprehensive information and logical text structure. Thus, I suggest it can be published with only minor revisions done. For example: Fig.2, the frame in the figure is distorted; P13L460, “2.1.3.” should be modified into “2.1.2.”

Reviewer 3 Report

The topic and timeliness of this review are superb, as there is a need for more efficient, modular, and diverse sensors. Without a doubt a combination of BODIPY dyes, whose versatility is unsurpassed, with various types of nanoparticles, does provide novel and unique sensors.

However, in the present form, the review is a mere recitation of the published accounts, without much attention to general trends, critical comparison among various accounts, etc. End of the section summaries (i.e., “to sum up” points) are helpful, but they are too general. Therefore, some additional section within each chapter, with critical, comparative analysis of the studies, results, structures, etc would significantly enhance the value of this review.

In addition, the authors might want to address in more detail the structural aspects of the BODIPY labels, with specific attention to specific functionalities. In other words, what type of functional groups are required for conjugation, sensing, etc. since it seems that BODIPY-core is largely preserved (with some variations: aza-BODIPY, styryl-containing, i.e., IR-range BODIPYs).

Maybe the authors would also like to consider a summary table, where both BODIPY, functional groups for conjugations and nanomaterials are organized in some manner.

Minor issues to consider:

1.    Fig. 17: how’s BODIPY conjugated to Au-NP (i.e.,, through which group)? Or is it just an adsorption? More details would be relevant.

2.    Fig. 20: figure and figure caption should have more information (what is “m” for example?), as a figure should be self-sufficient.

3.    Fig. 21: caption needs more details. Is orange line a dicaboxylate?

4.    Fig. 22: figure caption does not seem to be complete.

5.    Line 742: “instruments” – probes?

6.    Fig. 25: what is the evidence that the conjugation is taking place via NMe2 and not NEt2 (or both)?

7.    References are not uniformly formatted: some titles (e.g., 5, 15, 17, 21, 29, 34, etc) have every word capitalized, most only the 1stone.

Round 2

Reviewer 3 Report

The authors address the concerns, and the manuscript should be considered for publication.